**A high-resolution pan-Arctic meltwater discharge dataset from**

2                                    **1950 to 2021**

Adam Igneczi[1*], Jonathan Bamber[1,2]
[1]Bristol Glaciology Centre, School of Geographical Sciences, University of Bristol, UK
[2]Department of Aerospace and Geodesy, Technical University of Munich, Germany
* *Correspondance: Ádám Ignéczi <a.igneczi@bristol.ac.uk, ignecziadam@gmail.com>*
**Abstract**
Arctic air temperatures have increased about four times faster than the global average since
about 1980. Consequently, the Greenland Ice Sheet has lost about twice as much ice as the
Antarctic Ice Sheet between 2003 and 2019, and mass loss from glaciers and ice caps is also
dominated by those that lie in the Arctic. Thus, Arctic land ice loss is currently a major
contributor to global sea level rise. This increasing freshwater flux into the Arctic and North
Atlantic oceans, will also impact physical, chemical and biological processes across a range of
domains and spatiotemporal scales. To date, meltwater discharge data at Arctic coastlines
are only available from two datasets that are limited by their spatial resolution and/or
coverage. Here, we extend previous work and provide a high-resolution coastal meltwater
discharge data product that covers all Arctic regions, where land ice is present, i.e. the
Canadian Arctic Archipelago, Greenland, Iceland, Svalbard, Russian Arctic Islands. Coastal
meltwater discharge data – i.e. spatially integrated runoff that is assigned to the outflow
points of drainage basins – were derived from Modèle Atmosphérique Régional (MAR) daily
ice and land runoff products between 1950 and 2021, which we statistically downscaled from
their original ~6 km resolution to 250 m. The complete data processing algorithm, including
downscaling, is fully documented and relies on open-source software. The coastal discharge
database is disseminated in easily accessible and storage efficient netCDF files.

## 1. Introduction

Arctic air temperatures have increased about four times faster than the global average during the last four decades (Rantanen et al., 2022). One of the consequences of this is increasing land ice loss. The Greenland Ice Sheet (GrIS) has lost about twice as much mass as the Antarctic Ice Sheet between 2003 and 2019 (Smith et al., 2020, IPCC, 2021). Over the same period, glaciers and ice caps (GIC) in the Arctic – i.e. in Alaska, Canadian Arctic Archipelago, Iceland, Svalbard, Russian Arctic Islands – and peripheral GIC (PGIC) in Greenland were responsible for about 71% of the global GIC mass loss (Hugonnet et al., 2021). Altogether the GrIS and Arctic GIC lost a similar amount of ice during the last two decades. The rate of land ice loss has also been reported to have accelerated across the Arctic, except for Iceland (Ciracì et al., 2020), over the last few decades. Notably, mass loss rate in Greenland – i.e. the ice sheet and its PGICs – has been estimated to have increased sixfold between 1980 and 2020 (Mouginot et al., 2019). Due to these processes, Arctic land ice loss is currently a major contributor to global sea level rise (Frederikse et al., 2020; IPCC, 2021) and to the freshwater budget of the Arctic and North Atlantic oceans (Bamber et al, 2018).

Arctic GIC and the GrIS lose mass through a combination of decreasing surface mass balance – i.e. increasing surface runoff relative to precipitation – and increasing solid ice discharge (hereafter termed discharge). Although about two-thirds of the net mass loss from the GrIS between 1972-2018 is attributable to discharge (Mouginot et al., 2019), the relative contribution of this process has diminished to about 30-50% since 2000 due to increasing surface runoff (Enderlin et al., 2014; van den Broeke et al., 2016; Mouginot et al., 2019; King et al., 2020). This process plays an even more prominent role in land ice loss elsewhere in the Arctic; about 87% of the GIC mass loss between 2000 and 2017 across the Canadian Arctic Archipelago, Iceland, Svalbard, and the Russian Arctic Islands has been attributed to decreasing surface mass balance (Tepes et al., 2021). These trends illustrate the growing role of liquid meltwater discharge into Arctic seas, impacting physical, chemical and biological processes across a range of domains and spatiotemporal scales (Catania et al., 2020). Meltwater discharge at the ice-ocean interface of tidewater glaciers can also modulate discharge by influencing calving rates and ice dynamics (e.g.: Cowton et al., 2019; Melton et al., 2022). However, perhaps most importantly, increasing glacial freshwater flux – consisting of meltwater discharge and solid ice discharge – can influence the large-scale oceanic

circulation of the Arctic and sub-polar North Atlantic (SNA) Oceans (e.g.: Boning et al., 2016;
Gillard et al., 2016; Yang et al., 2016; Dukhovskoy et al., 2019; Biastoch et al., 2021) and
potentially the Arctic climate (Proshutinsky et al., 2015).
Despite its importance for a wide range of processes at varying spatiotemporal
scales, only two studies provide data covering a multi-decadal time span over most, but not
all, of Arctic land ice. These datasets rely on Regional Climate Model (RCM) runoff products –
Modèle Atmosphérique Régional (MAR) and/or Regional Atmospheric Climate Model
(RACMO) – digital elevation models, ice masks, statistical downscaling and meltwater routing
algorithms to estimate coastal surface runoff fluxes by reporting spatially integrated runoff at
coastal outflow points. Bamber et al. (2018) utilise RACMO2.3p2 and RACMO2.3p1 products
(1958-2016) – for the GrIS and GIC respectively – downscaled from 11 km to 1 km and cover
most of the Arctic and Sub-polar North Atlantic (SNA) Oceans region with significant land ice
presence, except for the Russian Arctic Islands. Although the coverage is fairly
comprehensive, the data is reported at a relatively low spatial (5 km) and temporal (monthly)
resolution. Mankoff et al., (2020) use both RACMO and MAR products (1950-2021) to provide
high resolution data – daily, with modelled runoff inputs downscaled from 7.5 km (MAR) and
5.5 km (RACMO) to 1 km and routed by using a 100 m resolution DEM – but only for
Greenland. Here, we attempt to combine the advantages of these two datasets, i.e. the high
resolution of Mankoff et al. (2020) and the large coverage of Bamber et al (2018), and provide
a high resolution (daily, downscaled to- and routed at 250 m) meltwater discharge dataset for
the period of 1950-2021. Our database is publicly available, efficiently stored – i.e. by
reporting runoff that is spatially integrated over drainage basins – and covers the most
important land ice sectors of the Arctic and SNA Ocean regions, i.e. the Canadian Arctic
Archipelago, Greenland, Iceland, Svalbard, Russian Arctic Islands.

## 2. An overview of the data processing pipeline

Our goal is to obtain a high resolution coastal meltwater discharge product that partitions meltwater according to its source, i.e. tundra, ice surface, and ice surface below the snowline (i.e. bare ice). To achieve this, we first downscaled coarse resolution (~ 6 km) RCM products: ice and tundra runoff, ice albedo; using their native vertical gradients and high resolution (250 m) surface DEMs (Figure 1). Downscaled ice albedo is only used to provide contextual information, i.e. to partition downscaled ice runoff according to its source (above or below the snowline). Limitations due to coarse resolution ice and land masks supplied with the RCM were addressed during this step by integrating high-resolution (250 m) ice and land masks into the downscaling algorithm (Figure 1). The high-resolution surface DEM that is used in the downscaling process is also used to delineate drainage basins and coastal outflow points in a hydrological routing algorithm. These drainage basins are used to sum the daily meltwater runoff and estimate meltwater discharge at the corresponding coastal outflow points (Figure 1). In order to limit computational requirements needed at any one time, we carried out the above process separately for each major glacier region. These are delineated according to the first order regions defined in in the Randolph Glacier Inventory v.6.0 (RGI Consortium, 2017): RGI-03 (Arctic Canada North), RGI-04 (Arctic Canada South), RGI-05 (Greenland), RGI-06 (Iceland), RGI-07 (Svalbard and Jan Mayen), RGI-09 (Russian Arctic) (Figure 2).

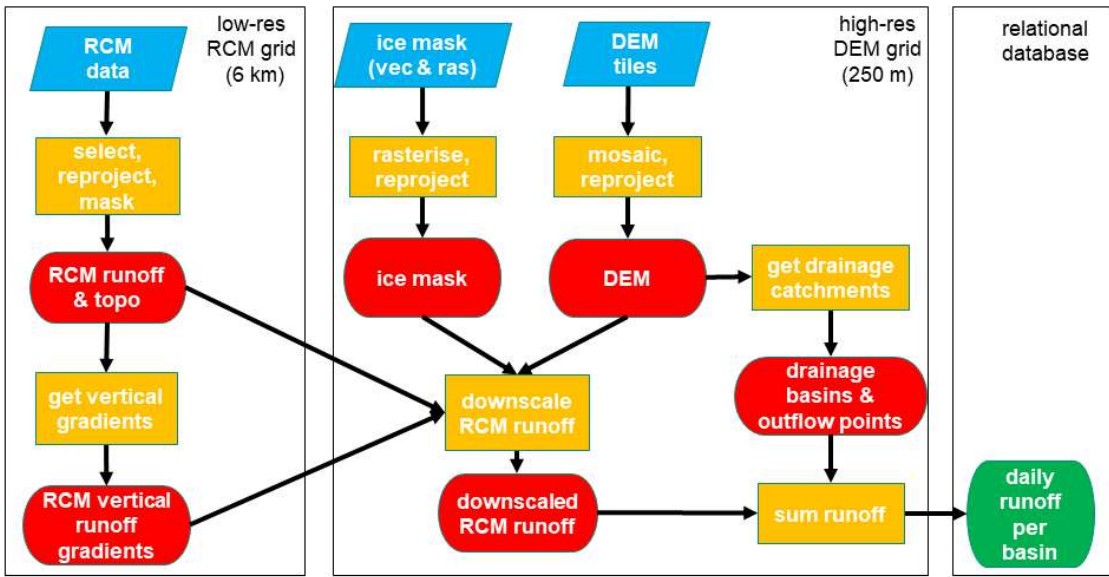

100

**Figure 1.** Data pipeline

## 3. Input data pre-processing

### 3.1. Static data

We assumed that time dependent changes in surface topography, land and ice extent have negligible impact on large-scale surface runoff during our period of interest, i.e. between 1950-2021. Hence we used static data products to obtain information about these physical properties.

### 3.1.1. DEM and land-ocean mask

High resolution (3"; ~90 m) DEMs were obtained from the Copernicus GLO-90 DGED DEM product (ESA, 2021). This DEM is distributed in 1°x1° tiles and is referenced on the WGS-84 ellipsoid. This product is in several ways superior to ArcticDEM – unless very high resolution (i.e. up to 1 m) is required – as it is gapless and resolves small islands and coastal areas precisely. ArcticDEM often has large elevation errors and significant data gaps close to coastal areas and small islands (e.g. Mankoff et al., 2020). Water Body Mask (WBM) tiles are also supplied with the GLO-90 DEM on the same grid. This provides a convenient way of separating terrestrial and oceanic domains which are consistent with the DEM. We used this product to create a binary land mask by selecting non-ocean pixels.

Using the RGI first order region outlines and the GLO-90 DEM grid shapefile we have
selected the required DEM and WBM tiles for each of the investigated RGI regions using the
open-source GIS software package QGIS. These tile lists, saved as text files, were used to
create DEM and WBM virtual mosaic files in the python geospatial library GDAL. After defining
the binary land-ocean masks from the WBM mosaics, we discarded DEM pixels coinciding
with the ocean mask to ensure we only retain valid DEM heights for terrestrial areas. The
mosaics were then reprojected in GDAL – using bilinear interpolation for DEM and nearest-
neighbour for WBM – to a 250 m grid referenced in an equal-area projected coordinate
system (North Pole Lambert Azimuthal Equal-Area Atlantic; EPSG:3574) to avoid the need for
scaling corrections further down the data pipeline due to area distortions (Snyder, 1987;
Bamber et al., 2018; Mankoff et al., 2020). Finally, the reprojected DEM and land-ocean mask
mosaics were clipped with the RGI region outlines. Henceforth we will refer to these products
as COP-250 DEM and COP-250 Land Mask. These products are also used further down the
data pipeline as reference grids for snapping.
**3.1.2. Ice mask**
As the RGI only provides glacier shapefiles for Greenlandic PGICs, we have used two
sources for our regional ice masks. Outside of Greenland we used RGI v.6.0 glacier outlines
(RGI Consortium, 2017). These are supplied in shapefiles referenced on the WGS-84 ellipsoid.
The shapefiles were first reprojected to EPSG:3574 and then rasterised to our reference 250
m grid (i.e. COP-250 DEM grid) using GDAL tools (ogr2ogr, gdal_rasterize) – a grid cell was
considered ice covered if its centroid was within RGI ice cover polygons. The COP-250 Land
Mask was then applied to correct for any potential mismatches (i.e. masking out oceanic
pixels) between the RGI and Copernicus datasets.

|  | Ice area relative difference (%) | Tundra area relative difference (%) |
|---|---|---|
| RGI-3 Canada North | 0.379 | -0.123 |
| RGI-4 Canada South | -0.022 | 0.002 |
| RGI-5 Greenland | 0.224 | -1.115 |
| RGI-6 Iceland | 0.065 | 0.002 |
| RGI-7 Svalbard | 0.705 | -0.936 |
| RGI-9 Russian Arctic | 0.967 | -0.562 |


**Table 1.** Relative difference between the original and 250 m resampled ice and tundra
domain areas (original minus 250 m resolution version) for each investigated RGI region.

For the GrIS and Greenlandic PGICs we have used the GIMP v.1 ice mask product (Howat et al., 2014; 2017). This is supplied as a mosaic for Greenland at a 90 m resolution grid referenced in a polar stereographic projection system (NSIDC Sea Ice Polar Stereographic North; EPSG:3413). After reprojecting it in GDAL – using nearest neighbour interpolation – to the COP-250 DEM grid, which is using the equal area EPSG:3574 projected coordinate system, we applied the COP-250 Land Mask to mask out potential oceanic pixels. Converting shapefiles and 90 m binary masks to 250 m binary masks, may lead to area discrepancies. However, based on our comparisons, bulk area discrepancies remain within the ±1% range (Table 1).

## 3.2. RCM products

Meltwater runoff and ice albedo both exhibit highly dynamic changes with time, thus we obtained information on these properties from daily RCM outputs provided by MAR v3.11.5 simulations (Fettweis et al., 2013, 2017; Maure et al., 2023) that were forced by 6 hourly ERA5 reanalysis data between 1950 and 2021. This product was chosen as it provides data at relatively high spatial (~ 6 km) and temporal (daily) resolution for a large geographical area, that almost completely covers our region of interest in the Arctic (Section 2). Altogether, MAR data covers 6 Arctic RGI domains, though the MAR domain delineations do not follow RGI conventions. Thus, MAR is distributed for 4 domains: Canadian Arctic (covering RGI-03 and RGI-04), Greenland (covering RGI-05), Iceland (covering RGI-06), and Russian Arctic and Svalbard (covering RGI-07 and RGI-09) (Figure 2). Although the MAR domains only offer partial coverage for some of their corresponding RGI regions, ice covered areas fall almost completely within the MAR domains, with only a negligible amount of glaciers excluded (Figure 2). However, a significant fraction of the tundra is not included in the RGI-03 (Arctic Canada North) and RGI-04 (Arctic Canada South) and to a lesser degree in the RGI-09 (Russian Arctic) regions (Figure 2). Thus, our data product cannot provide a full representation of the tundra runoff in these RGI regions. Incomplete coverage was also taken into consideration when delineating our drainage basins (Section 4.1) and when comparing our results with previous studies (Section 5.4).

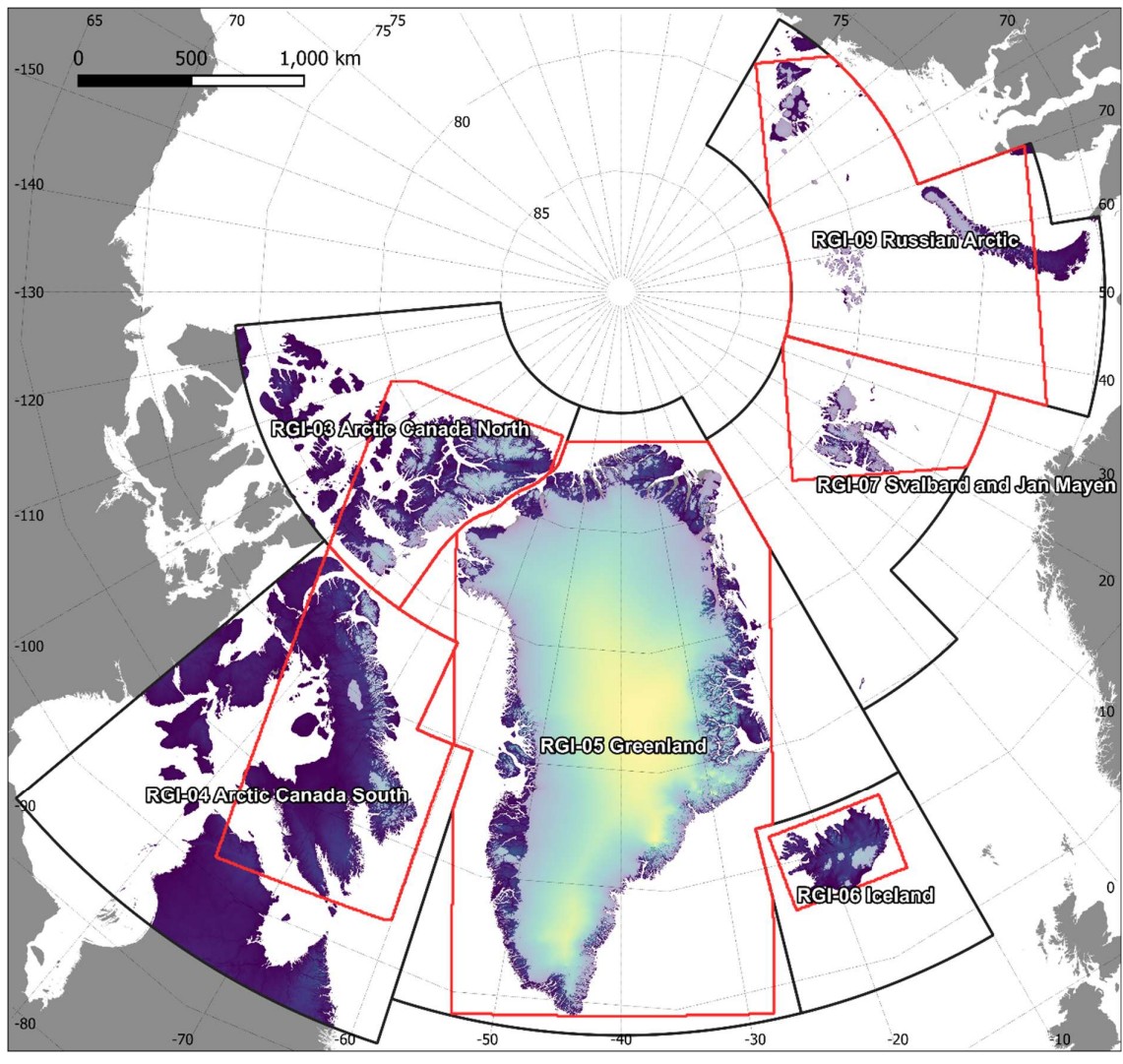

**Figure 2.** Overview map of our study area showing the COP-250 DEM with the ice coverage overlain (light shading). The investigated principal RGI regions (black line) and the MAR coverage (red line) are both displayed. MAR coverage plotted on the map has been clipped with the appropriate RGI region boundary.

MAR products are supplied in netCDF files, with each file holding a year's worth of daily data for a single MAR domain (i.e. there are 72 files for each of the 4 MAR domains). As the files contain many variables, we only extracted those we needed for our calculations (ice runoff, land/tundra runoff, ice albedo, surface elevation, and ice mask) to save computational time. Runoff, R, is defined as

$$R = ME + RA - RT - RF \text{ (Eq. 1)}$$

where ME is melt, RA is rainfall, RT is retention, and RF is refreezing. For tundra runoff RT and RF are both zero.

In lieu of a binary ice mask, this version of MAR introduces fractional ice coverage. Hence,
both land runoff and ice runoff data are provided for pixels with partial ice/tundra coverage.
The mask also contains generous fringe areas, where ice or tundra coverage is limited (< 0.001
%) and uniform. We simplified these fringe pixels by assuming them to be completely covered
by either ice or tundra. The corresponding ice or land runoff values were discarded (i.e. were
set to NoData), e.g. a pixel with 0.001% tundra coverage was assumed to be completely
covered by ice, thus the corresponding tundra runoff was discarded and ice runoff was
assumed to be valid for the whole pixel. This step reduced bias around ice-tundra boundaries,
e.g. during reprojection and resampling, and the calculation of vertical gradients.

MAR is referenced in a custom stereographic projection system, with a different set

of projection parameters for each domain. In addition, there is a 10° rotation for the Arctic
Canada domain, which needs to be reversed before reprojection. All MAR products were
reprojected from their custom system to EPSG:3574, while retaining their native 6 km
resolution. The reprojected MAR data were then clipped with the appropriate RGI region
boundary; this step also brings the MAR domains in line with the RGI regions thereby
consolidating our input data. During this step, we also saved the overlapping area between
the RGI regions and the MAR domains as shapefiles. This product is used further down the
processing pipeline to ensure that we are not extrapolating unreasonably beyond the spatial
coverage of valid MAR data. This issue, however, almost exclusively affects land runoff
products, as the ice covered regions within the investigated RGI regions are well captured by
MAR except for some small islands, e.g. Jan Mayen (Figure 2).

For computational efficiency, we have set up a parallel multiprocessing pool in

Python for each of the 6 investigated RGI region, with a dictionary ensuring that the
appropriate MAR domain is grabbed during processing. Then, we looped through the 72 years
covered by the MAR dataset and submitted each year separately to the pool as an
asynchronous task. Altogether 432 tasks were submitted, though the number of active
processes and pools were limited due to memory and core number constraints.

## 4. Methods

### 4.1. Drainage basins and outflow points

To obtain meltwater discharge volumes at Arctic coastlines, the RCM downscaling procedure needs to be combined with a hydrological routing scheme, which can use either the surface hydraulic head or the subglacial pressure head. In contrast to Mankoff et al. (2020), who assumed meltwater is immediately transported to the bed where it follows the subglacial pressure head, we have opted for a simpler approach and used surface routing exclusively. The principal reason for this is the lack of a pan-Arctic ice thickness product of sufficient accuracy and the relatively large uncertainty in bed topography even over the GrIS. Although, ice thickness estimates are available for all the RGI glaciers (Millan et al., 2022), this dataset is heavily reliant on shallow-ice approximation modelling and only covers Greenlandic PGICs and not the main ice sheet. The BedMachine product, which is based on mass conservation algorithms, is available for the latter region (Morlighem et al., 2017). However, ice thickness, especially for smaller glaciers outside Greenland, is highly uncertain compared to surface elevation. Furthermore, the aforementioned two datasets rely on fundamentally different methodology which would reduce the consistency of our input data.

The other source of uncertainty inherent to subglacial meltwater routing is due to the complexity of determining the exact timing, location and efficiency of surface-to-bed runoff capture. Although, it is well established that ice surface runoff can penetrate to the bed through ice of arbitrary thickness due to hydrofracturing (Das et al. 2008, Krawczynski et al., 2009), various factor influence this process, e.g. ice surface roughness, the pattern of surface fractures/crevasses, runoff volume, snow/firn thickness and saturation (Igneczi et al., 2018; Davison et al., 2019; Lu et al., 2021). Thus, meltwater can be routed for considerable distances on the ice surface before subglacial capture or proglacial discharge. Accordingly, supraglacial rivers exceeding several dozens of km-s in length, with some terminating at the ice margin, have been observed on the Devon and Barnes Ice Caps and in northern Greenland (Yang et al., 2019; Zhang et al., 2023). Connected to this issue, subglacial pressure head calculations usually assume that subglacial water pressure always equals the ice overburden pressure, i.e. the flotation-factor is constantly 1 (e.g. Mankoff et al., 2020). However, this assumption also introduces uncertainties as it disregards the spatiotemporal evolution of the subglacial drainage system (Davison et al., 2019).

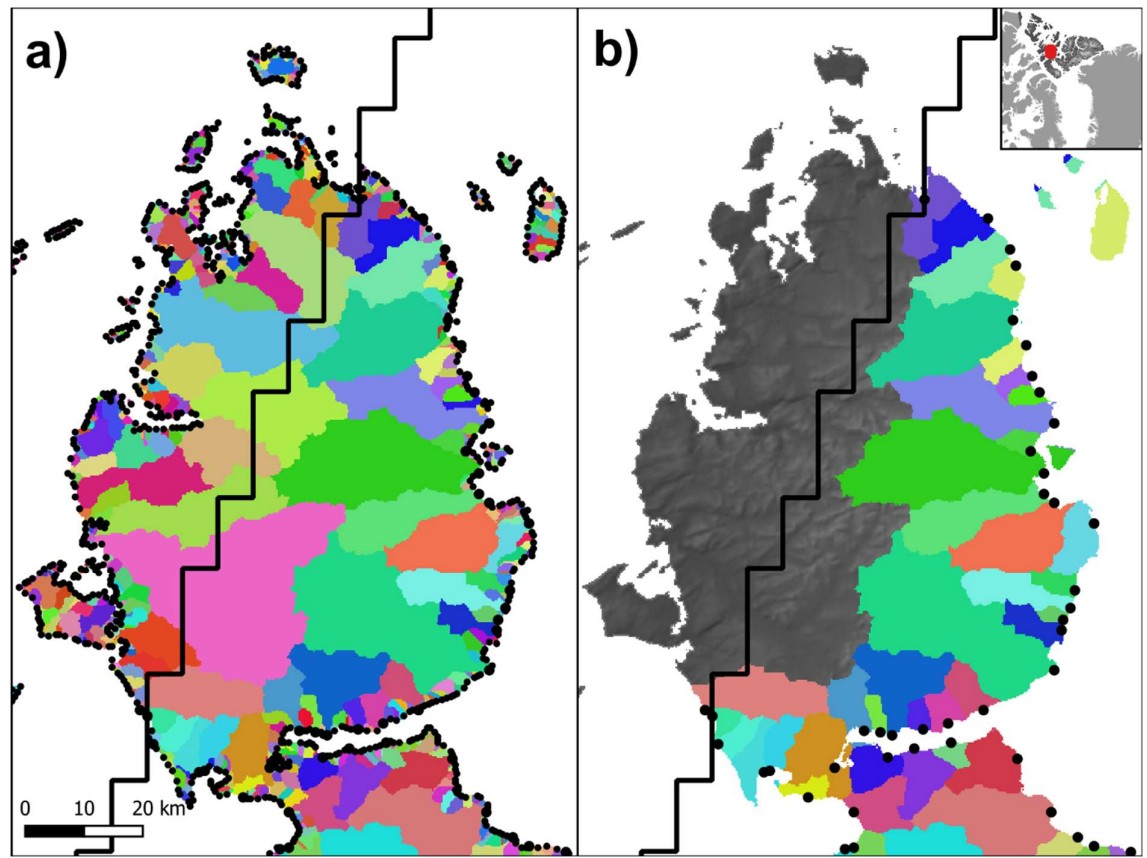

**Figure 3.** Surface drainage basins and their outflow points (black points) in Northern Canada
(a) before and (b) after the removal of small basins and basins that have at least 90% of their
area outside the MAR domain (solid black line).

In order to avoid these pitfalls and simplify our approach we used the previously

created COP-250 DEM product (Section 3.1.1) to calculate surface drainage basins. These
drainage basins were subsequently used to integrate the downscaled daily surface runoff
following the approach of Mankoff et al., (2020). The workflow is fully automated by using
the Whitebox tools (WBT) package in a Python script. After filling closed depressions and
treating flat areas – to ensure these have an outflow point – in the COP-250 DEM with the
*wbt.fill_depressions* tool (with the fix_flats option checked true), single D8 flow directions
were calculated using *wbt.d8_pointer*. Then, distinct drainage basins were derived from the
flow directions raster using the *wbt.basins* tool. The resulting product is an integer raster,
with unique integers indicating basin coverage (Figure 3). In order to limit the number of
basins, thereby aggregating our end product, we removed small basins ($< 10$ km$^2$) and set
their corresponding pixels to NoData. Then, we allocated these pixels to their nearest valid
basin using the *wbt.euclidean_allocation* tool (Figure 3). As this tool also assigns oceanic
pixels, we introduced an additional step to mask out the ocean. We also removed basins that
are touching the RGI region outline, buffered with the resolution of the COP-250 DEM. This
step ensures that all the drainage basins fall completely within the RGI domain. Data gaps in
the RCM products are filled in during the downscaling procedure to facilitate complete spatial
coverage (Section 4.3). However, to limit unreasonable spatial extrapolation, beyond the
coverage of MAR, we only retained surface drainage basins that have at least 90% of their
area within the MAR domain (Figure 3). Thus, altogether, 1.01%, 2.68%, and 3.85% of the
terrestrial MAR domain was discarded in Arctic Canada North, Russian Arctic, Arctic Canada
South, respectively. Other regions were unaffected by this step, and the discarded area had
negligible ice coverage.

Outflow points of the basins were calculated by finding pixels that have no flow

direction, i.e. no lower neighbours. These pixels were then converted to vector points and
saved to a shapefile. As the COP-250 DEM has previously been treated with the
*wbt.fill_depressions* tool with the fix_flat option – which ensures there are no closed
depressions and flat areas without outflow points, i.e. all pixels have a lower neighbour apart
from the edge pixels – these points will represent actual outflow points at the edges of the
basins. However, this step also yields the outflow point of basins that have been removed due
to their size or coverage (Figure 3). We have sampled the intermediate basin rasters to
identify and remove the outflow points that correspond to these removed basins. Thus, the
final product has a single outflow point for each valid basin, which is the outflow point
associated with the principal basin where fragments from smaller basins are included (Figure

3).

## 4.2. Vertical gradients of runoff and ice albedo

Localised regression analysis between elevation and modelled climatic parameters

has been used in various studies to statistically downscale reanalysis temperatures (e.g.:
Hanna et al., 2005; 2008; 2011; Gao et al., 2012; 2017; Dutra et al., 2020) and RCM estimates
of SMB components (e.g.: Franco et al., 2012; Noël et al., 2016, Tedesco et al., 2023). The
procedure of Franco et al., (2012) – downscaling MAR from 25 km to 15 km – relied on
localised vertical gradients that were obtained by calculating differences in elevation and
MAR variables within an 8-neighbourhood (8-N) moving window. They also applied vertical
weighing, i.e. averaged the vertical gradients by the total elevation difference within the
kernel, to dampen the influence of "extreme" local gradients. Noël et al. (2016) combined
elevation dependent downscaling – relying on localised linear regressions within a moving
window – with empirical accumulation, ablation, and bare ice albedo corrections. Tedesco et
al., (2023) relied solely on elevation dependent downscaling, which was carried out in a
similar manner to Noël et al. (2016) though SMB mass conservation was enforced within each
original MAR pixel. They also deployed a novel computational setup that achieved high
efficiency and speed by strongly leveraging parallelisation, which was enabled by highly
segmenting the input data.
All these studies – at their core – rely on the inherent localised vertical lapse rates of
RCM products. Thus, we have adopted a similar approach that utilises these lapse rates to
statistically downscale daily MAR products from their native resolution of ~6 km to the 250 m
resolution COP-250 DEM grid. The setup of our downscaling procedure is based on Franco et
al. (2012) due to its relative simplicity, i.e. relying on differences within the moving window
instead of linear regression. However, the elevation dependent downscaling carried out by
Noël et al. (2016) and Tedesco et al. (2023) is also similar – except for their use of linear
regression, additional empirical corrections, and mass conservation enforcement.
To calculate the required vertical gradients, first, an 8-N moving window was applied
to calculate the difference in elevation (i.e. the native DEM in MAR), ice runoff, land runoff,
and ice albedo – the latter for contextual purposes – between each pixel and their 8
neighbours. Ice and land runoff were handled separately to prevent "leakage" due to large
runoff contrast at the ice-tundra interface. Then, 8D local vertical gradients were determined
within the kernel by dividing ice runoff, land runoff, and ice albedo differences with their
corresponding elevation differences (Franco et al., 2012). NoData was assigned to the centre
of the kernel and 0 was assigned to every direction where the elevation difference is below
50 m, the latter step corrects for bias caused by elevation independent runoff and albedo
variance. This step is a substitute for vertical weighing (Franco et al., 2012) as it allows us to
filter out elevation independent variance – e.g. differences in runoff near the equilibrium line
due to the contrasting albedo and retention of snow/firn and bare ice – more completely and
precisely.
To yield local vertical gradient rasters, the average of the kernel gradients was
assigned to each central pixel if at least 5 valid gradients were found within the kernel.
Otherwise, the central pixel was assigned NoData. In lieu of carrying out our own sensitivity
analysis, we relied on the conclusions of Noël et al. (2016) who ascertained that using 6
regressions points – i.e. equivalent to 5 valid gradients – provides the best balance between
converging to, or diverging from the low resolution RCM runoff products. Positive vertical
gradients in ice/land runoff (i.e. runoff increasing with elevation) and negative vertical
gradients in ice albedo (i.e. ice albedo decreasing with elevation) were discarded, i.e. assigned
NoData. Data gaps were filled in using bilinear interpolation inside the convex hull of valid
data, and nearest neighbour extrapolation outside of it.

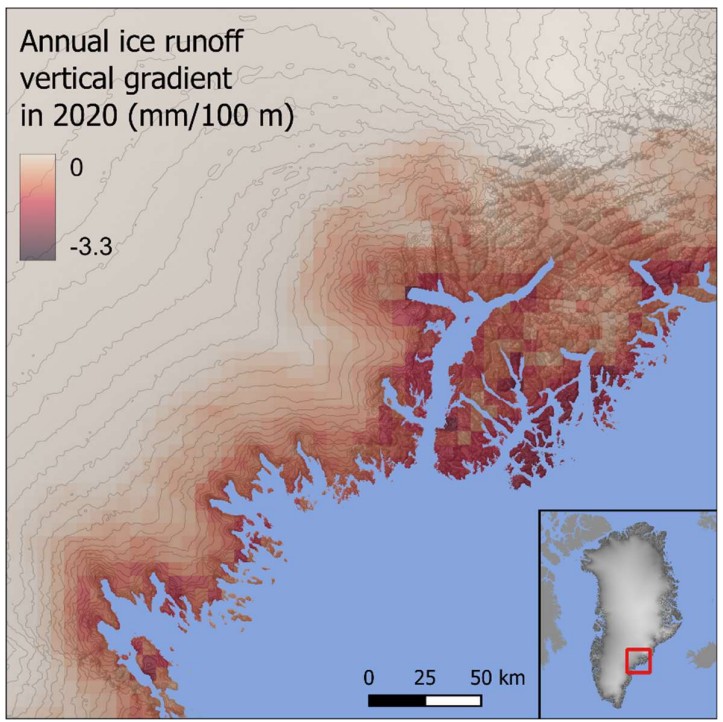


**Figure 4.** Annual average vertical ice runoff gradient for 2020 in SE Greenland; elevation
contours are drawn every 100 m. The annual average is calculated from the daily vertical ice
runoff gradients. Units are in mm/100 m, i.e. showing how many mm-s runoff will change
with every 100 m elevation gain.

To accurately track the temporal evolution of the vertical gradients, we sequentially

looped through each day covered by the MAR products. Thus, the process was carried out
26,298 times for each of the 6 RGI domains, producing 473,364 rasters with 6km resolution.
Annual time-averaged vertical gradients were also produced and saved to GeoTiffs for
reference (Figure 4). To save computational time, the task was integrated with the script that
carries out MAR pre-processing (Section 3.2). This design, in addition to taking advantage of
an already existing parallel processing scheme, facilitated efficient I/O operations by writing
pre-processed (i.e. filtered, reprojected, clipped) MAR products and their derived localised
vertical gradients to the same file – RGI domain specific yearly netCDF files – at the same time.
Although parallelisation was not leveraged as effectively as by Tedesco et al. (2023), the task
completed pan-Arctic pre-processing in about a day.

## 4.3. Statistically downscaled runoff and ice albedo

The first step of the statistical downscaling algorithm was upsampling the pre-

processed MAR ice, and tundra runoff, ice albedo (Section 3.2), their vertical gradients
(Section 4.2), and the MAR DEM from their native resolution of ~6 km to the 250 m resolution
COP-250 DEM grid. Nearest neighbour interpolation was first applied to fill in data gaps, then
upsampling to the COP-250 DEM grid was carried out by bilinear interpolation (Figure 5, 6,
S1). Once all products were upsampled to the COP-250 DEM grid, elevation differences were
calculated between the MAR DEM and the COP-250 DEM (Figure 5, 6, S1). Elevation
corrections were then made by multiplying the elevation difference with the appropriate
localised vertical gradient raster and adding this to the upsampled ice, and tundra runoff and
ice albedo rasters (Franco et al., 2012). Similar to the calculation of the vertical gradients, ice
and tundra runoff were handled separately to prevent biases caused by the high runoff
contrast at the ice-tundra interface. Henceforth we refer to these rasters as the downscaled
products. Oceanic pixels were masked out from all of the downscaled rasters by using the
high-resolution COP-250 Land Mask; while ice and tundra runoff were masked by the
appropriate high-resolution RGI or GIMP ice mask (Figure 5, 6, S1). Pixels with negative runoff
were assigned zero.

The downscaling procedure was carried out on the pre-processed daily MAR data,

which includes vertical gradients (Section 4.2). Although this procedure was handled
separately from MAR preprocessing, the computational setup is similar. A parallel
multiprocessing pool was created for each RGI region, then each task running asynchronously
on these pools grabbed a single year of data from the appropriate RGI region for processing.
Archiving downscaled daily runoff data – which have 250 m spatial resolution – would require
excessive storage capacity. To circumvent this problem, we only retained downscaled daily
runoff that was summed for the drainage basins. Thus, the algorithm, handling the integration
of runoff for the drainage basins (Section 4.4.), was combined with the downscaling
procedure. Annual runoff was also obtained for reference by summing the downscaled daily
products; these annual rasters were saved to GeoTiffs. Due to their large size, these files are
not published, but they are available on request.

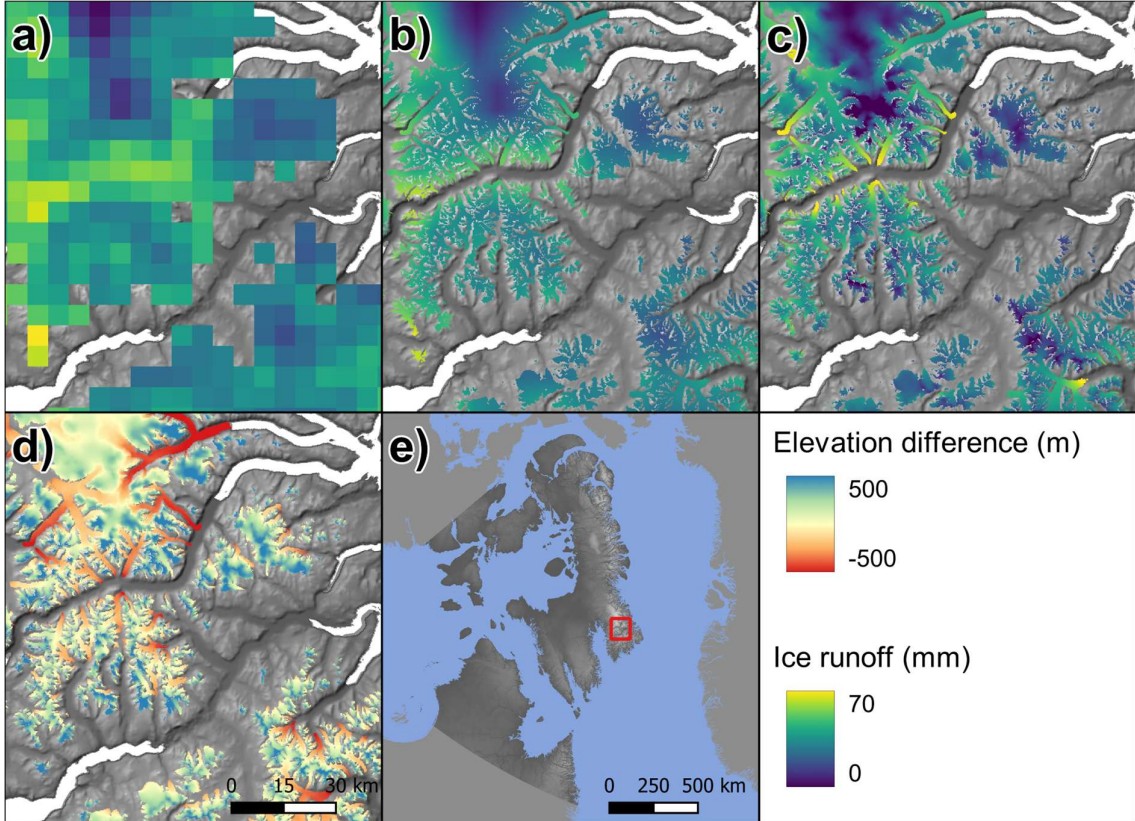


**Figure 5.** (a) Native resolution daily cumulative ice runoff for 19/July/2021 in Arctic Canada
South from MAR, runoff is plotted where fractional ice pixels indicate any amount of ice
coverage; (b) ice runoff after upsampling to 250 m; (c) ice runoff after elevation correction,
i.e. downscaling. (d) COP-250 DEM minus the upsampled MAR DEM within the RGI ice mask.
(e) Overview map.

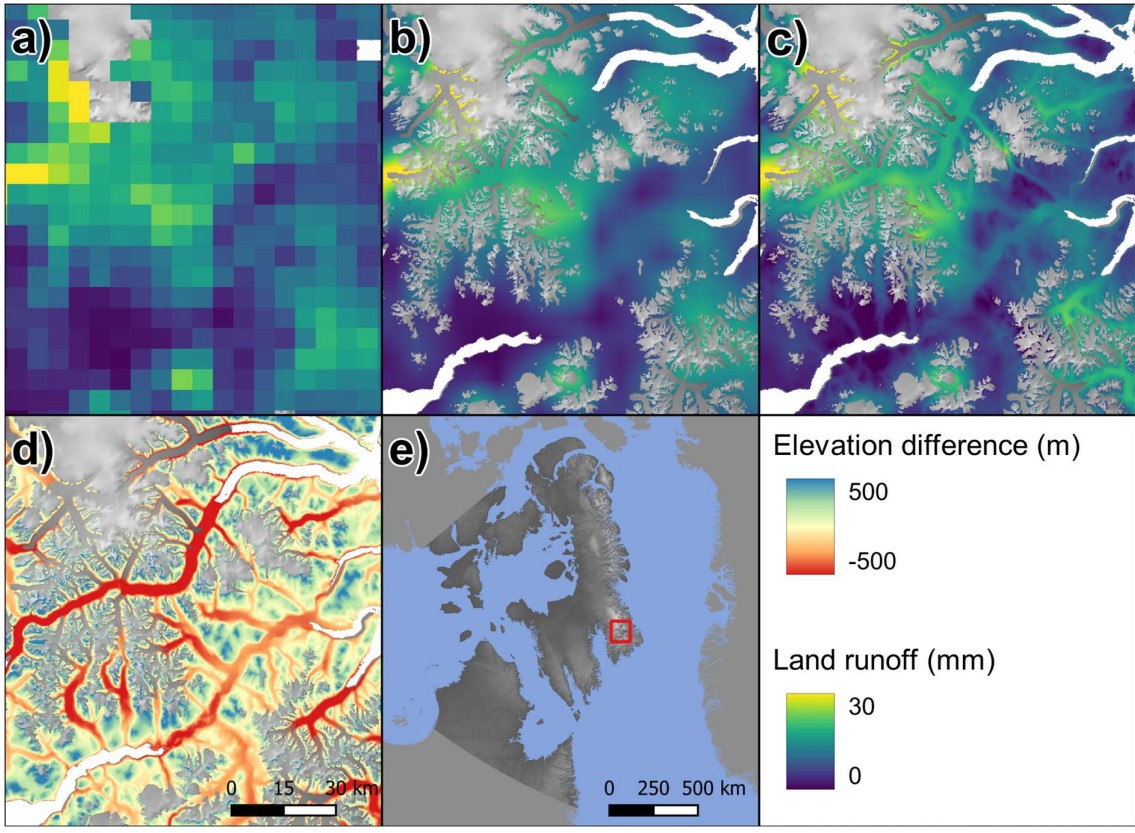

**Figure 6.** (a) Native resolution daily cumulative tundra runoff for 19/July/2021 in Arctic Canada South from MAR, runoff is plotted where fractional tundra pixels indicate any amount of tundra coverage; (b) tundra runoff after upsampling to 250 m; (c) tundra runoff after elevation correction, i.e. downscaling. (d) COP-250 DEM minus the upsampled MAR DEM outside the RGI ice mask. (e) Overview map.

Although our statistical downscaling procedure is similar to the one that was applied on the input data of Mankoff et al. (2020), there are several key methodological differences. Mankoff et al. (2020) used RCM products that have been downscaled to 1 km resolution – following the procedure of Noël et al. (2016) – prior to their data processing, i.e. statistical downscaling was not integrated into their routing algorithm. As the two procedures were separate, the resolution of their routing products (100 m) do not align with the resolution of their downscaled RCM products (1 km), and ice domains do not overlap precisely. To alleviate these spatial discrepancies, Mankoff et al. (2020) scaled and snapped RCM products to the routing resolution. Pixels with mismatching domain types (e.g. land according to RCM but ice according to the routing product) were assigned the average runoff of the corresponding ice/land basin. No runoff was reported for small basins with no RCM coverage of the same

type. As we carried out both the downscaling and the routing on the same grid, similar
adjustments were not needed in our data processing algorithm.

## 4.4. Meltwater discharge at outflow points

After downscaling, daily ice and land runoff was summed over each drainage basin.
In addition to carrying out this step for whole drainage basins, we also summed ice runoff
separately for subsections of the basins where the ice albedo was below 0.7. As this is the
minimum allowed albedo for the snow model in MAR (Fettweis et al., 2017), we propose that
runoff originating from these regions is a good approximation for runoff from below the snow
line (BSL). The reason for making this distinction is that, runoff above the snow line will be
predominantly due to melt of seasonal snow, while runoff BSL is predominantly ice and firn
melt and therefore a reduction in the "ice reservoir". This is an approximation but may be
useful for investigating secular versus seasonal fluxes. However, it is important to note that
MAR is known to overestimate bare ice areas, thus true snowline elevations might be lower
than estimated here (Ryan et al., 2019; Fettweis et al.., 2020).

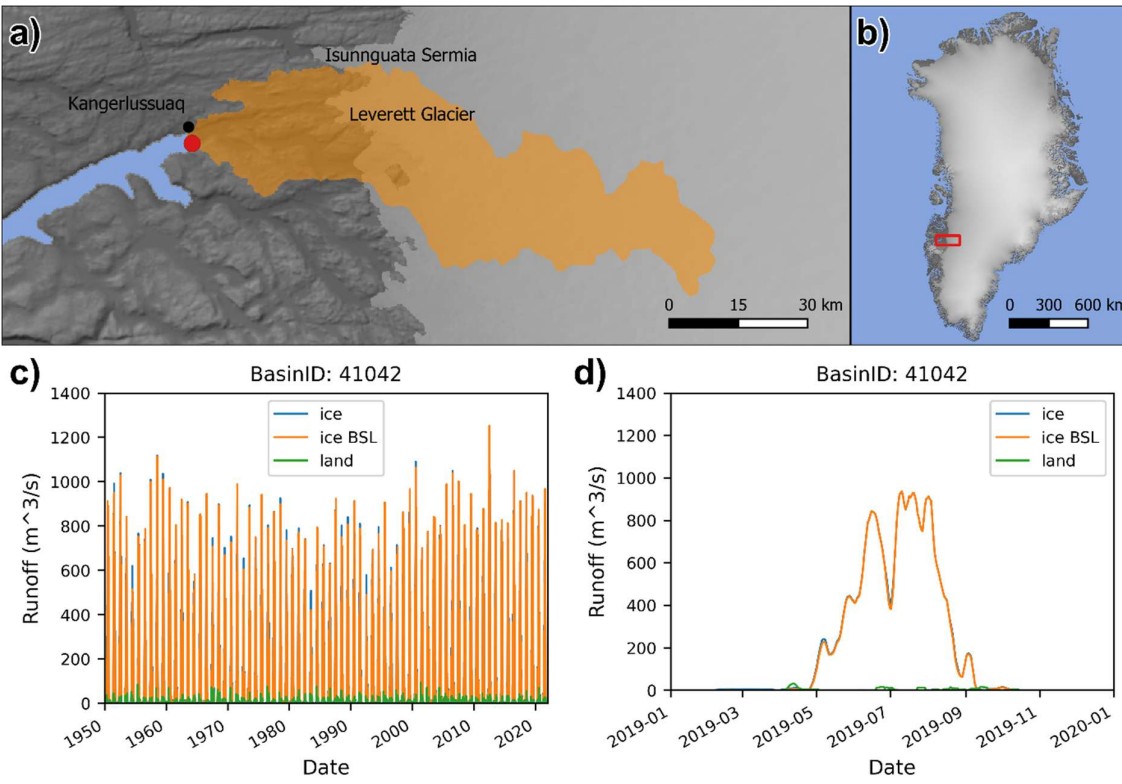


**Figure 7.** An example of our basin specific daily runoff data. (a) Coverage of the drainage basin,
which includes Leverett and Russel Glaciers in West Greenland, and its coastal outflow point,
(b) overview map. (c) Seven-day running average of the coastal meltwater discharge from ice,

land and bare ice – i.e. ice below snow line (BSL) – runoff between 1950 and 2021, (d) zoomed
in view of the same graph between 2019 and 2020.

The resulting basin specific daily runoff time-series were saved into three separate
tables – representing land, ice and bare ice runoff (Figure 7) – where rows represent days and
columns represent drainage basins. Due to the computational setup (Section 4.3), these
tables were initially saved to yearly RGI domain specific netCDF files. Thus, the final step was
concatenating these yearly files, to yield a single netCDF file for each RGI region which
contains the daily runoff data for each drainage basin within the region.

## 424   5. Product evaluation

### 425   5.1. Evaluation against river discharge measurements

To evaluate our product, we compared daily river discharge measurements from 7
locations in Greenland (Hawkings et al., 2016a, 2016b; Langley 2020; Sugiyama et al., 2014;
Kondo and Sugiyama 2020; van As et al., 2018) with our corresponding coastal meltwater
discharge time series, using the code published by Mankoff et al. (2020) for bulk comparisons.
Although river gauge data is available for 3 additional locations (Mankoff et al., 2020), we
were not able to integrate these with our product due to compatibility issues. Leverett Glacier
had to be removed as we only produce meltwater discharge time series at the coastlines, and
not at the glacier margins as in Mankoff et al. (2020). The four Greenland Ecosystem
Monitoring (GEM) river gauges near Nuuk – Kobbefjord, Oriartorfik, Kingigtorssuaq,
Teqinngalip – correspond to very small drainage basins, ranging from 7.56 to 37.52 $km^2$. Our
aggregation procedure – i.e. the merging of small basins (< 10 $km^2$) with their neighbours
(Section 4.1) – heavily affected these basins, thus direct comparisons with our products are
not possible. However, by investigating the topography and the non-aggregated basins of
Mankoff et al. (2020), we concluded that the neighbouring Kobbefjord and Oriartorfik gauges
– together – can reasonably represent discharge from the single aggregated basin that
contains them. Conversely, the Kingigtorssuaq and Teqinngalip gauges had to be completely
excluded as they only represent a small subsection of the aggregated basin that contains them
(Figure S2).

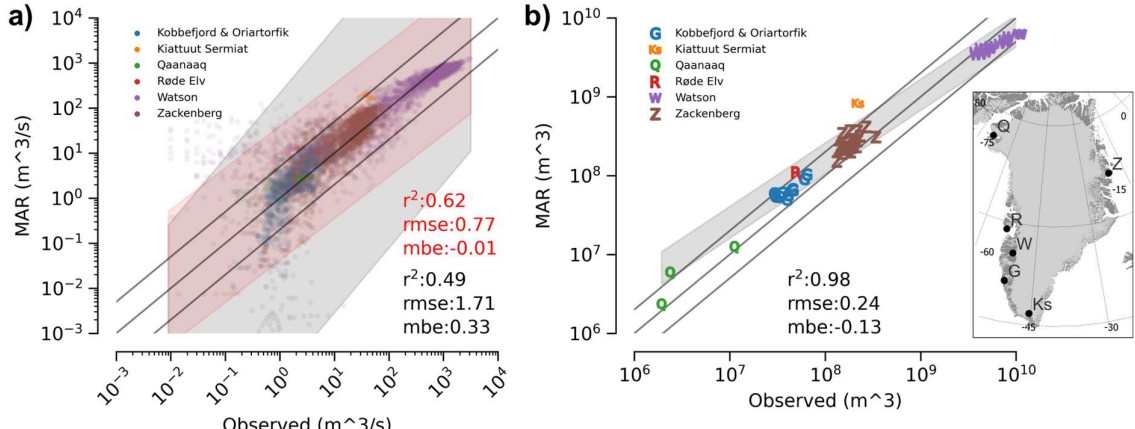


**Figure 8.** Bulk comparison of observed river gauge data and discharge derived from downscaled MAR. The map inset shows the location of the river gauges. Solid lines show 1:1 (centre), 1:5 (upper), and 5:1 (lower) correspondence. (a) Besides the original daily data, (b) annual sums calculated for calendar years are also compared. Grey band shows 5 % to 95 % prediction interval. Red band shows the same, when excluding the summed Kobbefjord & Oriartorfik data. $R^2$, root mean squared error (rmse), and mean bias error (mbe) are calculated after taking log10 of the data due to the huge value range. Drawn by utilising code from Mankoff et al. (2020).

Overall, the performance of our dataset against field measurements is very similar to the performance reported by Mankoff et al., (2020) for their MAR based discharge estimations. Both the $r^2$ values – 0.49, and 0.62 when excluding the GEM gauges near Nuuk – and the 5% to 95% prediction intervals of our daily data agree well with the equivalent results from Mankoff et al. (2020), who reported $r^2$ 0.45, and 0.59 respectively (Figure 8a). Our annual results – i.e. daily discharge summed by calendar year for the days when observations exist – also exhibit similar performance to Mankoff et al. (2020), who reported an $r^2$ of 0.96 which is close to our 0.98 (Figure 8b). However, our 5% to 95% prediction interval is slightly different. While the range is similar, it indicates that our dataset overestimates discharge towards the lower end of the annual discharge range; the negative mean bias error (-0.13) also confirms this overestimation. This is not surprising as we provide an aggregated product, i.e. very small basins are merged with their neighbours. The relative effect of the aggregation on discharge fidelity increases with decreasing basin size, which limits the feasibility of using our dataset for very small individual meltwater discharge outlets. However, it is important to note that bulk meltwater discharge is unaffected by this. Thus, we think the benefits of providing an aggregated product outweigh the limitations.

## 5.2. Comparison of downscaled and original MAR runoff

To reveal the specific effects of the downscaling procedure on our data product, we

compared bulk downscaled runoff with the original MAR runoff, separately for the ice and
tundra domains of each RGI region (Figure 9). Downscaled runoff and the original MAR runoff
exhibit characteristic differences that are largely independent of the runoff amount, i.e. vary
little year-to-year, and specific to each RGI region (Figure 9). This suggests that the factors
that determine the effect of downscaling on our runoff products are relatively static, and
inherent to the investigated regions. In general, downscaled ice runoff tends to be smaller
than the original MAR runoff (Figure 9, Table 2). This effect is the strongest in Arctic Canada
South and North (-23.5% and -12.5% respectively), elsewhere it remains moderate (between
-4.4% and -9%), while in Greenland downscaled runoff is slightly higher than MAR runoff
(+2.4%). On the other hand, downscaled tundra runoff is higher than the original MAR runoff
in all the investigated regions. This is the most significant in Svalbard (+28%), elsewhere it
remains more moderate (< 12.6%).

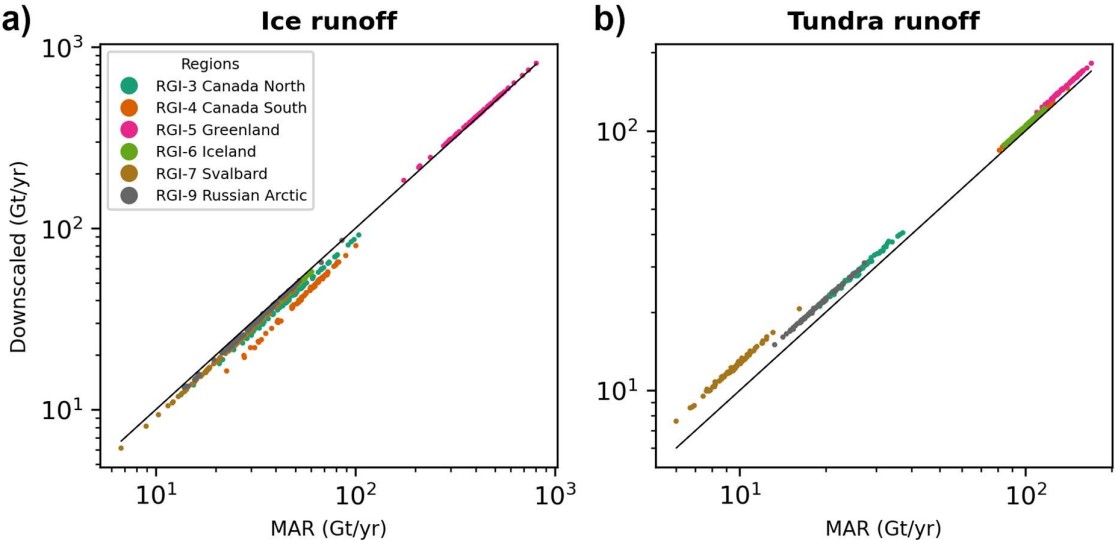


**Figure 9.** Annual sums of the original MAR runoff and the downscaled runoff, plotted
separately for (a) ice and (b) land areas of the investigated RGI regions.

Lower ice runoff in downscaled MAR mostly stems from reduction in ice area, due to

the differences between the MAR and high-resolution ice masks (Table 2). However, this is
not the only factor – e.g. in Greenland ice areas largely match, while ice area increases during
downscaling in the Russian Arctic (Table 2). Thus, topography, especially the difference
between MAR and high-resolution DEMs, also need to be considered. In general, the COP-250
DEM is lower than the MAR DEM within confined valleys, and higher along ridges, small

plateaus, and peaks; flat areas generally align well (Figure 5, 6, S3). If marine-terminating outlet glaciers – that drain ice from a flat interior all the way to the sea – dominate the glaciated landscape, then elevations are generally overestimated by MAR (Figure S4), and runoff will increase with downscaling. This effect has been pointed out for Greenland by several studies (e.g.: Bamber et al., 2001; Noël et al., 2016) and our results also align with it. However, if valley glaciers – which might terminate at higher elevations – smaller ice caps, and plateau glaciers dominate the landscape, then elevations are generally underestimated by MAR (Figure S4), and runoff will decrease with downscaling. This effect – along with the reduction in ice area – can reasonably explain why downscaling reduces ice runoff in Arctic areas outside Greenland.

| | Runoff RMSD | Runoff NRMSD (%) | Runoff average relative difference (%) | Area relative difference (%) |
|---|---|---|---|---|
| **Ice** | | | | |
| RGI-3 Canada North | 6.5 | 13.3 | -12.5 | -7.7 |
| RGI-4 Canada South | 13.3 | 23.4 | -23.5 | -16.6 |
| RGI-5 Greenland | 9.3 | 2.2 | 2.4 | -0.03 |
| RGI-6 Iceland | 2.4 | 5.5 | -5.5 | -5.0 |
| RGI-7 Svalbard | 2.2 | 9.1 | -8.7 | -6.9 |
| RGI-9 Russian Arctic | 1.3 | 4.2 | -4.4 | 3.7 |
| **Tundra** | | | | |
| RGI-3 Canada North | 2.9 | 10.9 | 10.8 | 4.4 |
| RGI-4 Canada South | 4.3 | 4.2 | 4.2 | 1.6 |
| RGI-5 Greenland | 10.1 | 7.3 | 7.3 | -0.4 |
| RGI-6 Iceland | 4.4 | 4.4 | 4.4 | 0.2 |
| RGI-7 Svalbard | 2.7 | 28.4 | 28.0 | 8.9 |
| RGI-9 Russian Arctic | 2.4 | 12.7 | 12.6 | -3.6 |

**Table 2.** Root Mean Squared Deviation (RMSD) was computed comparing the annual sums of the original and downscaled runoff, normalising (NRMSD) was carried by the annual sum of the original runoff. The average difference (downscaled minus original) was also normalised by the original MAR runoff. The difference in the domain area (high-resolution mask minus MAR mask) is also provided relative to the MAR domain area.

The increase in tundra runoff due to downscaling – when compared to the original MAR runoff – can also be connected to the reduction in ice area and the corresponding increase in land area during the downscaling procedure (Table 2). However, this relationship is not reciprocal as tundra area is also strongly influenced by the COP-250 Land Mask. Also, in some regions, tundra area decreases while the downscaled tundra runoff increases, e.g. in the Russian Arctic (Table 2). Thus, topography exerts a significant control on our tundra runoff

products too. In mountainous regions of the Arctic, tundra is typically situated at lower
elevations, e.g. the lower, non-glaciated sections of valleys – as the upper section of valleys,
higher ridges and plateaus are mostly glaciated. Thus, tundra elevations are often
overestimated by MAR, where confined valleys with non-glaciated lower sections are
abundant, e.g. in West Svalbard and South Novaya Zemlya (Figure S3, S5). Runoff will increase
with downscaling in such situations, which provides a good explanation for the observed
differences (Figure 9). However, further studies might be needed to fully uncover the
combined effect of such static factors and the complex spatiotemporal evolution of melting
on downscaling products.

## 5.3. Comparison with previous work

We also carried out bulk comparisons between our downscaled ice and tundra runoff
products and the equivalent datasets from Bamber et al. (2018) and Mankoff et al (2020).
Bamber et al. (2018) provide data for most of the Arctic (but less complete than here).
Conversely, the study of Mankoff et al. (2020) is restricted to Greenland. Thus, two sets of
comparisons were performed, one for Greenland and one for the rest of the Arctic. Runoff
products computed for the Russian Arctic were excluded from these comparisons, as this
region has not been investigated by either of the aforementioned two studies. As our MAR
domains – and thus our meltwater discharge dataset – only partially cover some RGI regions,
especially in Arctic Canada (Figure 2), and Bamber et al. (2018) provides more complete
coverage of the RGI domains, we clipped the Bamber et al. (2018) dataset with our MAR
domains (Figure 2). These steps ensured that the compared datasets have similar scope and
coverage.
Bulk ice runoff for Greenland agrees well between the three datasets. Although, the
1σ intervals of the three datasets – when comparing 5-year running means and standard
deviations – overlap well (Figure 10a), we estimated slightly larger runoff than the other two
datasets. The mean difference between our bulk ice runoff and that of Bamber et al. (2018)
and Mankoff et al. (2020) – when comparing datasets before applying running means – is
+17.7 Gt and +27.9 Gt (equivalent to +5.1 and +7.3% increase), respectively. Our estimation
for bulk ice runoff from glaciers and ice caps in other Arctic regions outside of Greenland
differs to a greater degree from the dataset of Bamber et al. (2018), i.e. with a mean
difference of +38.3 Gt (+40.1%) (Figure 10c). As bulk ice runoff only increases slightly in
Greenland (~2.4%) and decreases elsewhere due to our downscaling procedure (Section 5.2),
we propose that the differences in bulk ice runoff are mostly inherent to our MAR inputs. In
fact, downscaling brought our dataset more in-line with non-Greenland ice runoff products
of Bamber et al. (2018).

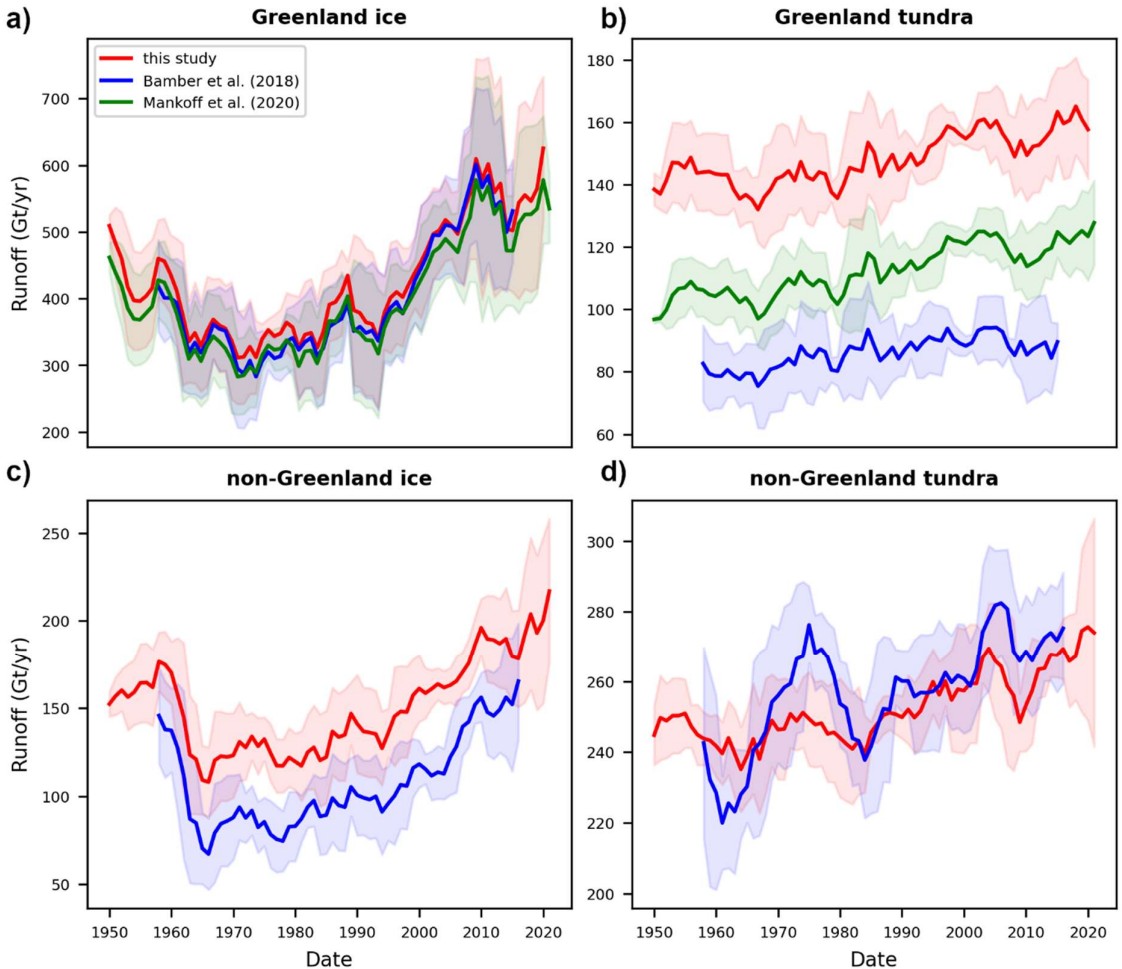


**Figure 10.** Bulk ice and land/tundra runoff for Greenland and all other Arctic regions, except
the Russian Arctic. Graphs show the 5-year running means, while shaded areas show the 5-
year running standard deviation. Note that Greenland ice includes PGIC.

The offset between land/tundra runoff estimates from the three datasets for
Greenland is larger than for ice runoff – with the 1σ intervals largely not overlapping – though
the trends and variability are very similar (Figure 10b). The mean difference between our bulk
land runoff and that of Bamber et al. (2018) and Mankoff et al. (2020) is +61.5 Gt and +36.1
Gt (+72.7% and +32.5%), respectively. Although alignment of the trends and variability of
tundra runoff estimations outside of Greenland is relatively poor, especially before 1980,
runoff magnitudes are similar without a clear pattern of over- or underestimation (Figure
10d). The mean difference between our product and the Bamber et al. (2018) dataset is -5.6
Gt (-1.7%), while the Root Mean Squared Deviation is 16 Gt. We believe the relatively poor
alignment of our non-Greenland tundra runoff pre-1980 with the Bamber et al. (2018) dataset
is related to their use of different RACMO versions in Greenland and the rest of the Arctic
(2.3p2 and 2.3p1 respectively) and the two sources of re-analysis forcings, ERA40 (1958-1978)
and ERA-Interim (1979-2016). Bulk tundra runoff increases everywhere in the Arctic due to
our downscaling procedure (Section 5.2). However, this increase is moderate in Greenland
(7.3 %), so only a fraction of the observed bulk runoff difference can be attributed to
downscaling. For non-Greenland tundra, where bulk runoff from the two products is similar
in magnitude, downscaling reduced inherent differences.
In conclusion, we propose that differences between our bulk ice and land runoff
results and the corresponding products by Bamber et al. (2018) and Mankoff et al. (2020), are
mostly inherent to our MAR inputs. As the three datasets differ substantially, it is difficult to
precisely explain the source of these inherent differences, however, different RCMs (MAR vs.
RACMO), different model versions (MAR 3.11 vs. MAR 3.11.5), different static (e.g. DEM and
ice mask) and dynamic (e.g. re-analysis) RCM forcings could be the most important factors.
Our downscaling procedure only played a secondary role, by reinforcing inherent differences
in Greenland and dampening them elsewhere. The exact reasons behind this warrant further
study.

## 578     6. Sources of uncertainty

Uncertainties have affected our products at various stages of processing. Firstly,
MAR products have introduced a degree of uncertainty into our results due to the physical
simplifications of the MAR model (e.g. Fettweis, 2020). Although MAR does not provide
formal spatiotemporally varying uncertainty products; based on analysis from the Greenland
Surface Mass Balance Intercomparison Project (GrSMBIP), its overall runoff uncertainty is
approximately ±15% (Fettweis et al., 2020).
The statistical downscaling procedure – which includes corrections applied to the
low-resolution MAR ice and land masks – has also introduced uncertainty into our runoff
products. Formal uncertainty that is specific to runoff downscaling is difficult to estimate as
localized in-situ runoff measurements are extremely sparse. Given this limitation, previous
investigations evaluated downscaled SMB estimations against in-situ measurements
collected in the field and found that downscaling reduced the RMSE by 9-24% in the ablation
zone (Noël et al., 2016; Tedesco et al., 2023). Although, these results are not directly
applicable to our study – as they refer to SMB, used different data sources, and applied
downscaling techniques that are somewhat different – they indicate that elevation
dependent downscaling can improve data quality. This, together with the validation and
comparison exercises we carried out (Section 5), suggest that the uncertainty profile of our
dataset is similar to previous products (e.g. Mankoff et al. (2020). We, therefore, consider our
product an improvement in terms of spatial coverage (compared to Mankoff et al., 2020) and
resolution (compared to Bamber et al., 2018), but not in terms of predictive performance
which remains in-line with previous products.
The final, coastal meltwater discharge product also has uncertainties due to the
simplified hydrological routing procedure. The first of these is caused by the assumption that
meltwater is routed on the surface. Meltwater can, and usually does, enter the englacial and
subglacial drainage system, where it follows a different hydraulic head. However, it is
complicated to quantify the location, timing and magnitude of subglacial capture, and the
exact path this meltwater follows. Therefore, it is difficult to ascertain which approach
introduces a larger uncertainty, using surface or subglacial routing exclusively. We have
mitigated this uncertainty by providing meltwater discharge only at the coastlines. This
implicitly carries out spatial averaging in areas where hydrological routing is only affected by
the surface hydraulic head, i.e. the location and magnitude of meltwater discharge at the ice-
land interface can be heavily affected by subglacial routing but this effect is weaker
downstream. However, this approach cannot mitigate uncertainty in ice-ocean discharge,
thus our product is less reliable at these interfaces.
The hydrological routing and the runoff integration procedure, has also assumed that
meltwater is instantaneously transported to the discharge point on the coastline. Besides the
actual transport time of meltwater within their conduits, which is affected by a complex array
of factors, many mechanisms can lead to meltwater retention and buffering (Forster et al.,
2014, Ran et al., 2024). MAR includes an approximation for retention and release of
meltwater in the firn layer, and a time delay for bare ice runoff (Fettweis et al., 2013, 2017;
Maure et al., 2023), though these are expected to be highly uncertain. Retention, storage,
and release of meltwater in the surface- (e.g. in supraglacial ponds, terrestrial lakes and
regolith), englacial/subglacial- (e.g.: in moulins, subglacial lakes, cavities, and sediment), and
proglacial hydrological system (e.g.: frontal and lateral lakes, lakes on the tundra,
groundwater) are completely unaccounted for. For instance, the duration of buffered
meltwater storage in the Greenland Ice Sheet can range between 4 and 9 weeks (Ran et al.,
2024). Thus, a significant delay can occur between melting and discharge at the coastal
outflow point. These factors introduce uncertainty into the estimated discharge volume time-
series at the coastlines.

## 6. Code and data availability

Data are available at https://doi.pangaea.de/10.1594/PANGAEA.967544 (Igneczi
and Bamber, 2024). Code is available at:
https://github.com/igneczidadam/meltwater_discharge.git

## Author contribution

AI and JLB were both involved in the conceptualization of the work. JLB was
responsible for funding acquisition. AI developed the methodology, carried out data curation,
software development, and visualization with contributions from JLB. AI prepared the original
draft, with review and editing contributions from JLB.

## Competing interests

The contact author has declared that none of the authors has any competing interests

## Acknowledgements

This work was funded by the European Union's Horizon 2020 research and
innovation programme through the project Arctic PASSION (grant number: 101003472). We
also thank X. Fettweis for providing MAR outputs. JLB was also funded by the German Federal
Ministry of Education and Research (BMBF) in the framework of the international future AI
lab "AI4EO -- Artificial Intelligence for Earth Observation: Reasoning, Uncertainties, Ethics and
Beyond" (grant number: 01DD20001).

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
