# Peer review of "A high-resolution pan-Arctic meltwater discharge dataset from"

_Earth System Science Data, 2024_

## Referee Comment (RC1)

**Review of *"A high-resolution pan-Arctic meltwater discharge dataset from 1950 to 2021"***
by Adam Igneczi et al.

The authors present long-term timeseries of daily pan-Arctic land ice and tundra runoff (1950-2021). Daily outputs from the regional climate model MAR at 6 km are first statistically downscaled to 250 m spatial resolution in sub-regions, i.e., Greenland, Canadian Arctic, Svalbard, Iceland and Russian Arctic, and further spatially integrated at the basin scale. The authors find that spatially integrated land ice and tundra runoff are larger than previous estimates at lower spatial resolution (pan-Arctic or Greenland only), while sharing overall similar variability and trends. The authors suggest that the larger runoff at 250 m stems from enhanced spatial resolution relative to previous products, the result of small glaciers and rugged tundra regions being better resolved.

While this data set will be of interest to the cryosphere community, I have major concerns regarding its evaluation. The authors claim that higher spatial resolution relative to previous estimates results in improved runoff representation. However, without a thorough data evaluation as e.g., in Mankoff et al. (2020) for the Greenland ice sheet, it is impossible to verify this statement. In addition, the actual impact of statistical downscaling on MAR integrated runoff is not assessed. Runoff increase relative to previous lower-resolution products could potentially originate from an overall runoff overestimate in the downscaled MAR product. This must be examined in more detail to support the authors claim that the presented data set is an improvement upon previous products. Based on the above and the following general and point comments, I deem that major revisions are required before considering this study for publication in ESSD.

**General comments**

1. Data evaluation is crucial to assess whether land ice and tundra runoff timeseries are robust, and an actual improvement upon previous products. To do so, the authors could use discharge measurements (e.g., for Greenland rivers in Mankoff et al., 2020) and modelled runoff estimates from e.g., (statistically downscaled) regional climate models that have been thoroughly evaluated in previous publications (i.e., using in-situ and remote sensing data). Such data sets exist for most ice masses in the Arctic, but a thorough evaluation for the well-studied Greenland ice sheet would be highly beneficial.

2. The authors do not assess the impact of their statistical downscaling technique, i.e., how does downscaling MAR at 6 km to 250 m affect integrated runoff in different regions? This is particularly important for smaller Arctic ice masses that may not be well resolved in low-resolution MAR.

3. The authors claim that higher spatial resolution improves runoff representation based on a previous study (Noël et al., 2016). However the latter work uses a different regional climate model combined with a different statistical downscaling technique, which does not imply that similar improvements will hold for MAR. For instance, Tedesco et al. (2023) found better agreement with in-situ surface mass balance measurements in Greenland after statistically downscaling MAR at 6 km to 100 m using a mass conservative approach (i.e., no runoff increase between resolutions). Model evaluation is therefore essential to ensure that enhanced runoff at 250 m found in this study does not reflect an overall overestimate in downscaled MAR.

4. The MAR version used in this study is never mentioned. This is important information for cross-study comparisons e.g., is it the same MAR version as in Mankoff et al. (2020) and/or Tedesco et al. (2023)?

5. The authors should elaborate on the difference between their statistical downscaling method and that of e.g., Franco et al. (2012), Noël et al. (2016), and Tedesco et al. (2023).

6. The paper and its Figures are mostly focused on Greenland. It would be beneficial to show downscaled outputs from smaller glaciers, to provide insights on the technique performance in different regions.

**Point comments**

L1: The authors could consider "basin scale" instead of "high-resolution" in the title.

L16: I am concerned about using the term "improve" as no detailed model evaluation is performed. I strongly recommend evaluating the presented data set with existing (high-resolution) runoff products

(modelled and observed) to show that the larger runoff found in this study compared to previous ones e.g., Bamber et al. (2018) and Mankoff et al. (2020), is an actual improvement.

L16: I suggest "basin scale meltwater discharge data product".

L19: "Modèle Atmosphérique Régional (MAR)".

L67-68: The authors state that the higher spatial resolution in this study (250 m) improves runoff estimates (L16, L374-377, L392-395). However, the data set in Mankoff et al. (2020) is available at 100 m. Please clarify.

L71: Do the authors mean "1950-2021", the year 2022 is not shown or discussed elsewhere.

L79: As downscaling uses elevation gradients, this implies that a relationship between MAR ice albedo and surface elevation exists. Is this the case? How well does MAR at 6 km represent ice albedo and the location of the snowline? This is important as the location of the snowline, and hence the bare ice extent and associated albedo, have a strong impact on the modelled runoff amount and spatial distribution. For instance, Ryan et al. (2019) showed that MAR overestimates bare ice extent in Greenland (and potentially the runoff production?).

L124: After interpolating the GIMP/RGI ice masks onto the 250 m grid, how did the authors cope with fractional ice cover? Did you discard e.g., grid-cells with <50% of ice coverage? Please elaborate. Do the resulting ice/land mask area align with observations, especially for smaller glaciers?

L140: Please specify which MAR version is used here.

L141: Do the authors mean "six hourly"?

L145-146: I count six MAR domains in Figure 2. The Svalbard region is not mentioned. Same comment in L155 "4 domains".

L146-149: What about the tundra area? Are they well captured by MAR? How robust are runoff estimates over the tundra regions?

L172-174: Why not using the original grid instead of interpolating MAR on a regular 6 km grid? This may lead to additional uncertainties. Please clarify.

L197-202: In fact, data mix has already been applied to create Greenland masks, i.e., combining Copernicus GLO-90 DGEDEM for surface elevation and land mask, GIMP DEM v1 for ice sheet mask, and RGI v6 for peripheral GIC mask. For instance, the GIMP DEM could have been used for surface elevation, land and (peripheral) ice masks. Please clarify or reformulate.

L237-238: How are these data gaps filled? Please clarify. Do the authors mean "Section 4.3"?

L238-240: How does discarding smaller basins affect the total land ice/tundra integrated areas?

L244: The paper refers to python tools and options that readers may not be familiar with. The authors could briefly explain what these consist of.

L255-258: The authors should elaborate on differences with previously published downscaling techniques, notably that of Tedesco et al. (2023) using MAR at 6 km as input. I am not sure to understand how downscaled ice albedo is used in the calculation of runoff.

L262: Are tundra and land ice runoff gradients computed separately? How do the authors downscale runoff at the interface between tundra and land ice?

L263-265: Discarding vertical gradients for elevation difference < 50 m may affect runoff production nearby the equilibrium line, i.e., towards the flatter glacier interior.

L269-271: This is valid for RACMO, but does it hold for MAR?

L281-285: What are the differences with previous downscaling techniques?

L319: I think Fig. 5 shows the opposite: "COP-250 DEM minus MAR DEM" with outlet glaciers (ice divides) elevation being overestimated (underestimated) in low-resolution MAR.

L337-343: The upper threshold for bare ice albedo is commonly set to 0.55 and is used to discriminate snow from ice. Could you please clarify why a snow value of 0.70 is used instead?

L362-364: I strongly recommend that the authors compare both the original and downscaled MAR runoff, i.e., to assess the impact of the statistical downscaling technique.

Section 4: Differences in Greenland ice sheet and peripheral GIC runoff can reach up to ~50 Gt between this study and the two previous products (Fig. 7a), i.e., 10 to 15% of the total. This is significant. For Greenland tundra and non-Greenland land ice, these differences are even larger. This calls for a proper data evaluation using discharge measurements (Mankoff et al., 2020) or previously published (high-resolution) land ice runoff products.

L374-377: This is somewhat speculative. The statement can only be verified by statistically downscaling MAR to different spatial resolutions and comparing the outputs with MAR at 6 km. In addition, the other data sets are based on a different regional climate model combined with different downscaling techniques, which may also explain the discrepancies.

L380-383: This statement suggests that the current data set outperforms that of Bamber et al. (2018). However, without proper evaluation, it is impossible to assess.

L383-384: Different reanalysis forcing will indeed impact the results. The input regional climate model will also strongly affect the results as they may not produce identical runoff amount and distribution.

L396-399: Trends and variability are similar for Greenland land ice, tundra and non-Greenland land ice, i.e., although with a positive runoff offset in the current study. However, it is not the case for non-Greenland tundra. Could you elaborate on this?

L400-403: I am not sure to understand how statistical downscaling in tundra regions is important for Greenland (L392-395) but not for other Arctic regions? Is the tundra region rougher in Greenland? Please explain.

L407: I am confused, I understood that MAR at 6 km was used as input. Was it formerly downscaled to higher resolution before applying your downscaling technique? Please clarify.

L411-413: I strongly recommend performing a thorough data evaluation, otherwise it is impossible to assess whether this new data set is robust, or an improvement upon other products.

L415-420: The fact that improvement was found in Noël et al. (2016) using RACMO, with a different downscaling technique, does not imply similar results when using MAR as input. For instance, Tedesco et al. (2023) suggest that mass conservative statistical downscaling (i.e., no runoff increase) is required to better align with in-situ measurements in Greenland.

L420-422: The variability and trends are mostly similar between products, but the runoff offset in this study remains important, calling for a proper model evaluation.

**Figures**
Fig. 2: Please add a scale bar for surface elevation.

**References**
Mankoff et al. (2020): https://essd.copernicus.org/articles/12/2811/2020/
Noël et al. (2016): https://tc.copernicus.org/articles/10/2361/2016/tc-10-2361-2016.html
Tedesco et al. (2023): https://tc.copernicus.org/articles/17/5061/2023/
Franco et al. (2012): https://tc.copernicus.org/articles/6/695/2012/
Bamber et al. (2018): https://iopscience.iop.org/article/10.1088/1748-9326/aac2f0/meta
Ryan et al. (2019): https://www.science.org/doi/10.1126/sciadv.aav3738

---

## Author Comment (AC1)

**Review of *"A high-resolution pan-Arctic meltwater discharge dataset from 1950 to 2021"***

by Adam Igneczi et al.

The authors present long-term timeseries of daily pan-Arctic land ice and tundra runoff (1950-2021). Daily outputs from the regional climate model MAR at 6 km are first statistically downscaled to 250 m spatial resolution in sub-regions, i.e., Greenland, Canadian Arctic, Svalbard, Iceland and Russian Arctic, and further spatially integrated at the basin scale. The authors find that spatially integrated land ice and tundra runoff are larger than previous estimates at lower spatial resolution (pan-Arctic or Greenland only), while sharing overall similar variability and trends. The authors suggest that the larger runoff at 250 m stems from enhanced spatial resolution relative to previous products, the result of small glaciers and rugged tundra regions being better resolved. While this data set will be of interest to the cryosphere community, I have major concerns regarding its evaluation. The authors claim that higher spatial resolution relative to previous estimates results in improved runoff representation. However, without a thorough data evaluation as e.g., in Mankoff et al. (2020) for the Greenland ice sheet, it is impossible to verify this statement. In addition, the actual impact of statistical downscaling on MAR integrated runoff is not assessed. Runoff increase relative to previous lower-resolution products could potentially originate from an overall runoff overestimate in the downscaled MAR product. This must be examined in more detail to support the authors claim that the presented data set is an improvement upon previous products. Based on the above and the following general and point comments, I deem that major revisions are required before considering this study for publication in ESSD.

Reply
Thank you for the thorough review. We have attempted to take on board all suggestions and criticisms and provided a substantially revised manuscript. Main additions include a validation against river gauge data, and comparisons of the original and downscaled runoff.

**General comments**

1. Data evaluation is crucial to assess whether land ice and tundra runoff timeseries are robust, and an actual improvement upon previous products. To do so, the authors could use discharge measurements (e.g., for Greenland rivers in Mankoff et al., 2020) and modelled runoff estimates from e.g., (statistically downscaled) regional climate models that have been thoroughly evaluated in previous publications (i.e., using in-situ and remote sensing data). Such data sets exist for most ice masses in the Arctic, but a thorough evaluation for the well studied Greenland ice sheet would be highly beneficial.

Reply
We entirely agree, although it is important to note here that we are not developing an SMB time series but a runoff one. As a consequence this limits some comparisons. We compared our coastal discharge estimations with Greenlandic river gauge data using code published by Mankoff et al. (2020), and the same dataset – though with some restrictions. A new section was written to explain the findings (Section 5.1). To summarise, the performance of our dataset is very similar to the MAR derived product of Mankoff et al. (2020). We have included    insights from this exercise and from the comparisons between our original and

downscaled runoff, and revised Section 5.3, which describes the comparisons against other downscaled RCM runoff products. Section 5 was also modified accordingly.

Based upon the new results from these additional evaluation steps we clarify our previous claim about "improvement" to the following: "We consider our product an improvement in terms of spatial coverage (compared to Mankoff et al., 2020) and resolution (compared to Bamber et al., 2018), but not predictive performance which remains in-line with previous products."

2. The authors do not assess the impact of their statistical downscaling technique, i.e., how does downscaling MAR at 6 km to 250 m affect integrated runoff in different regions? This is particularly important for smaller Arctic ice masses that may not be well resolved in low resolution MAR.

Reply
We compared our downscaled and original MAR runoff, separately for tundra and ice in each investigated RGI region. A new section is now included to explain the findings (Section 5.2). To summarise, bulk ice runoff slightly increases in Greenland due to downscaling (+2.4%) but decreases elsewhere (between -4.4 and -23.5%). Bulk tundra runoff increases due to downscaling in all regions (between 4.2 and 28%). We have also investigated the potential factors that could have influenced the net effect of downscaling on bulk runoff. We have found that differences in the MAR and high-resolution ice- and land masks, and DEMs, along with the topographical configuration of each region provide reasonable explanations. We reviewed Section 5.3 given these new insights. Overall, we propose that the observed differences between this study and previous products are not primarily caused by our downscaling procedure, as they are mostly inherent to the MAR inputs.

3. The authors claim that higher spatial resolution improves runoff representation based on a previous study (Noël et al., 2016). However the latter work uses a different regional climate model combined with a different statistical downscaling technique, which does not imply that similar improvements will hold for MAR. For instance, Tedesco et al. (2023) found better agreement with in-situ surface mass balance measurements in Greenland after statistically downscaling MAR at 6 km to 100 m using a mass conservative approach (i.e., no runoff increase between resolutions). Model evaluation is therefore essential to ensure that enhanced runoff at 250 m found in this study does not reflect an overall overestimate in downscaled MAR.

Reply
We agree that this argument is a generalisation and needs further investigation. It is often argued that high resolution downscaling resolves low lying valleys better and thus will lead to increased runoff (e.g. Bamber et al., 2001; Noël et al. 2016). Using comparisons between our original and downscaled runoff, we can support this argument for Greenland, though the net effect is fairly small (~2.4% overestimation). In fact, for GICs outside Greenland the opposite is true, i.e. downscaling leads to an underestimation of the original MAR runoff due to poorly represented ridges and plateaus. Thus, the enhanced ice runoff that we report does not originate from a downscaling overestimation. On the other hand, downscaling tends to overestimate tundra runoff in most regions. Thus, enhanced tundra runoff at least partly originates from downscaling.

Although, enforcing mass conservation (within each RCM pixel) is an interesting proposition, it is not standard procedure. We remain unconvinced that it is necessary to follow this approach, as several studies have shown that net changes in SMB are legitimate during downscaling (e.g. Noël et al., 2016; Noël et al., 2023). Reasons why runoff, in particular, is influenced by resolution are discussed and analysed quantitatively in Bamber et al, 2001 for example, indicating that mass conservation for runoff at least is an inappropriate assumption.

4. The MAR version used in this study is never mentioned. This is important information for cross-study comparisons e.g., is it the same MAR version as in Mankoff et al. (2020) and/or Tedesco et al. (2023)?

Reply
We did include this infromation in the original m/s but accept it could have been easily overlooked. We used MAR v 3.11.5 as stated in the manuscript (the version number is on a new line right before the citations). It slightly different than what Mankoff et al. (2020) used (3.11) and the same as     used by Tedesco et al. (2023).

5. The authors should elaborate on the difference between their statistical downscaling method and that of e.g., Franco et al. (2012), Noël et al. (2016), and Tedesco et al. (2023).

Reply
The setup of our downscaling procedure is based on Franco et al. (2012) due to its relative simplicity, i.e. relying on differences within the moving window instead of linear regression. However, the elevation dependent downscaling carried out by Noël et al. (2016) and Tedesco et al. (2023) is also similar – except for their use of linear regression, additional empirical corrections, and mass conservation enforcement.
Furthermore, we want to emphasize that it was not our intention to develop a novel methodology that has improved performance. Rather we aimed to build on existing methods and extend their usage, both by applying them on a larger scale and documenting the whole process (including both downscaling and routing).

We also edited the text of Section 4.2 to highlight the similarities and differences between our workflow and the work of Franco et al. (2012), Noël et al. (2016), and Tedesco et al. (2023).

6. The paper and its Figures are mostly focused on Greenland. It would be beneficial to show downscaled outputs from smaller glaciers, to provide insights on the technique performance in different regions.

Reply
We agree with this suggestion, so we have swapped West Greenland for Southern Arctic Canada on Figure 5, we also include a new figure (Figure 6) which shows tundra runoff downscaling for the same region. The original Figure 5 has been moved to the supplementary material (Figure S1).

**Point comments**

L1: The authors could consider "basin scale" instead of "high-resolution" in the title.

Reply
We appreciate the suggestion and agree that it could be an alternative term. However, we prefer to retain high-resolution as we believe it is an appropriate characterisation of our product. Also, we think it is much more accessible to readers, who might get confused by the term "basin-scale" without reading the methods first.

L16: I am concerned about using the term "improve" as no detailed model evaluation is performed. I strongly recommend evaluating the presented data set with existing (high-resolution) runoff products (modelled and observed) to show that the larger runoff found in this study compared to previous ones e.g., Bamber et al. (2018) and Mankoff et al. (2020), is an actual improvement.

Reply
Thank you for pointing out the significance of using the term 'improve'. We agree that to characterise our product as an improvement upon previous work would require a thorough model evaluation. Furthermore, we want to emphasize that it was not our intention to develop a novel methodology that has improved performance. Rather we aimed to build on existing methods (for both downscaling and routing) and extend their usage, both by applying them on a larger scale and documenting the whole process (including both downscaling and routing). We have modified the wording used accordingly.

Following previous comments by the reviewer, we have now included the outcomes of such an evaluation, which demonstrate similar performance to previous datasets. Given this outcome and our original research aim (explained above), we have edited the manuscript to reflect our intentions and outcomes better, e.g. we changed 'improve' to 'extend' in the abstract.

L16: I suggest "basin scale meltwater discharge data product".

Reply
Please see our previous reply.

L19: "Modèle Atmosphérique Régional (MAR)".

Reply
Corrected.

L67-68: The authors state that the higher spatial resolution in this study (250 m) improves runoff estimates (L16, L374-377, L392-395). However, the data set in Mankoff et al. (2020) is available at 100 m. Please clarify.

Reply
We meant the higher resolution of the downscaling, i.e. Mankoff et al. (2020) relied on MAR outputs that were downscaled to 1 km (prior to applying their routing workflow) whilst their routing was carried out at 100 m. In order to keep our data processing streamlined we have carried out the downscaling and the routing at the same resolution (250 m). Indeed, this is lower resolution than the routing done by Mankoff et al. (2020), but higher than their underlying runoff data. Thus, we believe that potential benefits due to higher resolution downscaling might be better realised in this study (e.g. due to better representation of low-lying narrow valleys where melt is the highest, see Bamber et al. 2001).

However, as proving "better" performance is not straightforward we modified our terminology related to this throughout the manuscript. Also, we edit the text here (L67-68) to make it clear that we are not considering our study an improvement on Mankoff et al (2020) in terms of resolution, rather we aimed to extend the spatial coverage.

L71: Do the authors mean "1950-2021", the year 2022 is not shown or discussed elsewhere.

Reply
That's correct, thank you for pointing out this mistake.

L79: As downscaling uses elevation gradients, this implies that a relationship between MAR ice albedo and surface elevation exists. Is this the case? How well does MAR at 6 km represent ice albedo and the location of the snowline? This is important as the location of the snowline, and hence the bare ice extent and associated albedo, have a strong impact on the modelled runoff amount and spatial distribution. For instance, Ryan et al. (2019) showed that MAR overestimates bare ice extent in Greenland (and potentially the runoff production?).

Reply
Thank you for raising this important point. We acknowledge that albedo variability is especially complex, and just relying on vertical gradients will not provide a precise downscaled product. This is the principal reason why we are not using the downscaled albedo product in any way to correct the downscaled runoff (e.g. similar to MODIS data used by Noël et al. 2016); it only provides contextual information (i.e. to estimate the ratio of runoff originating from below the snowline) that some users might find useful.

We also acknowledge that MAR albedo has uncertainties, though it has been thoroughly evaluated against other models and observations (e.g. Fettweis et al. 2020). As we only aim to locate the snowline, we are mostly affected by the CROCUS snow model formulations within MAR, which have their minimum albedo set to 0.7 (e.g Fettweis et al., 2013; 2017). Thus, we are less affected by bare ice albedo from MAR which is particularly uncertain.

To raise attention to these issues, we edit the text (here and throughout) to make it clear that ice albedo is only downscaled to provide contextual information. We also point the readers towards Ryan et al. (2019) in Section 4.4 to indicate that MAR overestimates bare ice extent (and thus snowlines might be located lower than what we estimated).

L124: After interpolating the GIMP/RGI ice masks onto the 250 m grid, how did the authors cope with fractional ice cover? Did you discard e.g., grid-cells with <50% of ice coverage? Please elaborate. Do the resulting ice/land mask area align with observations, especially for smaller glaciers?

Reply
To keep data processing streamlined we relied on simple nearest neighbour interpolation for GIMP re-gridding, similarly a grid cell was considered ice covered if its centroid was within RGI ice cover polygons. This simple procedure circumvented the need to consider fractional coverages and allowed us to consolidate all data inputs to the same resolution and grid, which makes our steps less complex. Although ice masks are less precise due to this simplification and their lower resolution, we found that their total area remained within ±1% of the original. Also, we found that 250 m provides a good balance between representing small glaciers and

limiting computational resources that are required to carry out the downscaling and routing procedure for the whole Arctic.

Additional description is provided at the relevant sections to explain the data processing more precisely.

L140: Please specify which MAR version is used here.

Reply
We used MAR v 3.11.5 as stated in the manuscript (the version number is on a new line right before the citations).

L141: Do the authors mean "six hourly"?

Reply
Yes, that is correct. Thank you for pointing out this omission.

L145-146: I count six MAR domains in Figure 2. The Svalbard region is not mentioned. Same comment in L155 "4 domains".

Reply
This is due to the mismatch between MAR domains and RGI domains.
This version of MAR is provided for four domains: Arctic Canada, Arctic Russia, Greenland, Iceland. Arctic Canada covers RGI-03-Arctic Canada North and RGI-04-Arctic Canada South. Arctic Russia covers RGI-07 Svalbard and RGI-09 Russian Arctic.

The MAR domain names in the manuscript follow their original naming conventions. However, I have altered this slightly (Arctic Russia ☐ Arctic Russia and Svalbard) to avoid misunderstandings.

Additional explanation has also been included to make the mismatch between MAR and RGI domains more clear.

L146-149: What about the tundra area? Are they well captured by MAR? How robust are runoff estimates over the tundra regions?

Reply
Thanks for pointing out this omission. Indeed, as Figure 2 shows MAR only provides partial coverage of tundra areas for Arctic Canada South and North and Arctic Russia. We now highlight this in the text, and point the readers to the relevant sections where we described how this issue is taken into consideration (when delineating drainage basins in Section 4.1 and when comparing our results with previous studies in Section 5.3).

L172-174: Why not using the original grid instead of interpolating MAR on a regular 6 km grid? This may lead to additional uncertainties. Please clarify.

Reply
In that case, we would have had to convert all other data (land and ice masks, and DEMs) to 4 different custom projection systems (not referenced in EPSG) which would also introduce uncertainties. Also, those stereographic projection systems do not preserve area, thus scaling

corrections would be necessary. We found it more straightforward to integrate MAR data with all other data sources into a single equal-area projection system.

L197-202: In fact, data mix has already been applied to create Greenland masks, i.e., combining Copernicus GLO-90 DGEDEM for surface elevation and land mask, GIMP DEM v1 for ice sheet mask, and RGI v6 for peripheral GIC mask. For instance, the GIMP DEM could have been used for surface elevation, land and (peripheral) ice masks. Please clarify or reformulate.

Reply
Thank you for raising our attention on this issue.
It is true that in contrast to all other RGI regions we cannot rely on the RGI v6 ice mask product to create a comprehensive ice mask for Greenland. That is the reason why we use the GIMP ice mask to represent both the ice sheet and PGICs, as stated in the 2$^{nd}$ paragraph of Section 3.1.2. Thus, we only introduce a data mix for ice masks (one source for Greenland and one for the rest of the Arctic), while using a single DEM source across the Arctic. If we relied solely on GIMP for Greenland, then we would have a data mix for both ice masks and DEMs across the Arctic. Additionally, the GIMP DEM is itself a data mix (relying on SPOT-5, AVHRR, and ASTER, thereby its "true" resolution varies between 40 and 500 m), whilst the Copernicus DEM is more consistent (relies on TanDEM-X and only uses other sources for local infilling). Thus, we prefer to rely on the Copernicus-Dem where possible.

The key reason why we avoided using ice thickness data (and thus using subglacial pressure heads for routing) is the high uncertainty of the data, especially for RGI glaciers, and the additional uncertainties (i.e. surface-to-bed capture, flotation-factor). The Millan et al. (2022) dataset also relies on fundamentally different methods than BedMachine, which we propose represents a more significant mismatch than what is between GIMP and RGI ice masks.

To make our reasoning clearer we now emphasize the uncertainty issues more than the data-mix issues.

L237-238: How are these data gaps filled? Please clarify. Do the authors mean "Section 4.3"?

Reply
Thank you for noticing this. Yes, we refer to Section 4.3 where we described the upsampling procedure, nearest neighbour interpolation at native 6 km resolution followed by bilinear interpolation to 250 m.

L238-240: How does discarding smaller basins affect the total land ice/tundra integrated areas?

Reply
Small basins (<10km$^2$) were not discarded, but merged with their largest neighbour, thus here is no effect on bulk runoff and total area.
L238-240 refers to drainage basins that have a fraction of their area outside the MAR domain. Regardless of the size of these basins, we removed them if more than 10% of their area lied outside the MAR domain (Figure 2 red outline), to reduce the scale extrapolation of MAR data.

This issue only affects tundra in Arctic Canada and Russia (as Iceland, Svalbard and Greenland are completely covered by MAR), and can cause the integrated area included in

our investigation (i.e. covered by drainage basins) to be either smaller (more usually) or larger than the area covered by MAR in the region. Please see the attached map and data for specifics.

[Figure]

Dissolved drainage basins outline (blue) is shown for the regions affected by the issue of basins covering areas without MAR data. The most significant effect is visible in the Arctic Canada South region (e.g. on the northern tip of the Labrador peninsula).

The relative effect on total integrated area is
Greenland: 0%
Svalbard: 0%
Iceland: 0%
Canada North: +1.01%
Canada South: -3.85%
Russian Arctic: -2.69%

L244: The paper refers to python tools and options that readers may not be familiar with. The authors could briefly explain what these consist of.

Reply
We included additional explanation about the WhiteBox hydrological tools that were utilised by our workflow.

L255-258: The authors should elaborate on differences with previously published downscaling techniques, notably that of Tedesco et al. (2023) using MAR at 6 km as input. I am not sure to understand how downscaled ice albedo is used in the calculation of runoff.

Reply
Downscaled ice albedo is not used in the calculation of runoff (only as contextual information to determine the proportion of the runoff that originates from below the snowline). Therefore, our procedure is very similar to Tedesco et al. (2023), though we are not applying mass conservation (also we downscale runoff instead of SMB). Our procedure is even more similar to the v.0.2 downscaling of Noël et al. (2016), i.e. downscaling using elevation dependence only (without applying ice albedo and precipitation corrections). The largest difference between our approach and these studies is in the way we calculate vertical gradients. Instead of carrying out local linear regressions using an 8-N moving window, we use the simpler approach of calculating differences in elevation and runoff using an 8-N moving window (as described in Section 4.2).

We have added a new sentence to Section 2 to state in advance that ice albedo is not used to adjust the downscaled runoff. We also edited the text of Section 4.2 to highlight the similarities and differences between our workflow and the approach of Franco et al., 2012, Noël et al. (2016) and Tedesco et al. (2023).

L262: Are tundra and land ice runoff gradients computed separately? How do the authors downscale runoff at the interface between tundra and land ice?

Reply
Yes, tundra and ice runoff were handled separately throughout the procedure (i.e. both during the calculation of vertical gradients and downscaling) to prevent "leakage". As explained in the 1$^{st}$ paragraph of Section 4.3 data gaps both in MAR runoff and vertical gradients (e.g. at the ice-tundra interface due to the low-resolution of the MAR mask) are filled by using nearest neighbour interpolation, before upsampling to 250 m by bilinear interpolation. After downscaling, the high-resolution land and ice masks are applied (see Figure 5, S1, S2).

L263-265: Discarding vertical gradients for elevation difference < 50 m may affect runoff production nearby the equilibrium line, i.e., towards the flatter glacier interior.

Reply
We agree with the reviewer, however we believe that it is appropriate to take steps to dampen the effect of elevation independent runoff variance on our elevation dependent downscaling exercise, e.g. Franco et al. (2012) also took such steps. We propose that it is reasonable to assume that other factors dominate runoff variance in flatter regions, e.g. near the equilibrium line this might be the difference between snow/firn /bare ice albedo. Also, low resolution RCM DEMs agree better with high resolution DEMs in flatter regions, which limits the impact of discarding vertical gradients there.

We included a sentence to provide additional explanation.

L269-271: This is valid for RACMO, but does it hold for MAR?

Reply
Although we haven't carried out the same sensitivity experiment, we investigated the daily difference between MAR and downscaled runoff (figure attached here), which yielded a trend that is very similar to the one demonstrated by Noël et al. (2016) (Figure 5 in their paper). Noël et al. (2016) also argue that – apart from the extremes, i.e. requiring 3 or 8 neighbours – the downscaled runoff is not particularly sensitive to this choice.

Tedesco et al. (2023) required 5 valid neighbouring pixels, when calculating vertical gradients, even though they have downscaled MAR instead of RACMO. Thus, we believe it is sensible to align our choice for this number with the findings of Noël et al. (2016) and the application of Tedesco et al. (2023).

[Figure]

Average daily (between 1950-2011) difference between downscaled (i.e. elevation corr.) and MAR ice runoff (downscaled minus MAR) for Greenland.

L281-285: What are the differences with previous downscaling techniques?

Reply
We added additional explanation (about two new paragraphs) to the beginning of Section 4.2 which describes previous downscaling approaches in detail and explains how our methods compare to them. We also described the computational setup of Tedesco et al. (2023) and explain how our parallelisation compares.

L319: I think Fig. 5 shows the opposite: "COP-250 DEM minus MAR DEM" with outlet glaciers (ice divides) elevation being overestimated (underestimated) in low-resolution MAR.

Reply
Thank you for pointing this out. Yes, it shows COP-250 DEM minus MAR DEM. I have edited the caption to make this more clear.

L337-343: The upper threshold for bare ice albedo is commonly set to 0.55 and is used to discriminate snow from ice. Could you please clarify why a snow value of 0.70 is used instead?

Reply
Thank you for pointing out this. We imprecisely refer to "runoff from bare ice areas" when in fact we mean runoff from below the snowline. This is the reason why we used 0.7 as the albedo threshold to find the snowline and partition runoff.

The text has been edited to reflect this better.

L362-364: I strongly recommend that the authors compare both the original and downscaled MAR runoff, i.e., to assess the impact of the statistical downscaling technique. Section 4: Differences in Greenland ice sheet and peripheral GIC runoff can reach up to ~50 Gt between this study and the two previous products (Fig. 7a), i.e., 10 to 15% of the total. This is significant. For Greenland tundra and non-Greenland land ice, these differences are even larger. This calls for a proper data evaluation using discharge measurements (Mankoff et al., 2020) or previously published (high-resolution) land ice runoff products.

Reply
We agree with the suggestion and undertaken two tasks to address it:
1.) We compared our coastal discharge estimations with Greenlandic river gauge data using code published by Mankoff et al. (2020), and the same dataset – though with some restrictions. A new section was written to explain the findings (Section 5.1). To summarise, the performance of our dataset is very similar to the MAR derived product of Mankoff et al. (2020).
2.) We also compared our downscaled and original MAR runoff, separately for tundra and ice in each investigated RGI region. A new section is now included to explain the findings (Section 5.2). To summarise, bulk ice runoff slightly increases in Greenland due to downscaling (+2.4%) but decreases elsewhere (between -4.4 and -23.5%). Bulk tundra runoff increases due to downscaling in all regions (between 4.2 and 28%). We have also investigated the potential factors that could have influenced the net effect of downscaling on bulk runoff. We have found that differences in the MAR and high-resolution ice- and land masks, and DEMs, along with the topographical configuration of each region provide reasonable explanations.

We reviewed Section 5.3 given these new insights. Overall, we propose that the observed differences between this study and previous products are not primarily caused by our downscaling procedure, as they are mostly inherent to the MAR inputs.

L374-377: This is somewhat speculative. The statement can only be verified by statistically downscaling MAR to different spatial resolutions and comparing the outputs with MAR at 6

km. In addition, the other data sets are based on a different regional climate model combined with different downscaling techniques, which may also explain the discrepancies.

Reply
We have now carried out the task of comparing our downscaled product with the original MAR runoff. According, to the findings of this comparison (detailed in Section 5.2 and in our reply to the previous comment) we have removed this statement. We also include the argument expressed in our reply to the previous comment to Section 5.3.

L380-383: This statement suggests that the current data set outperforms that of Bamber et al. (2018). However, without proper evaluation, it is impossible to assess.

Reply
We agree with the reviewer (please see in our previous replies) and revised this paragraph.

L383-384: Different reanalysis forcing will indeed impact the results. The input regional climate model will also strongly affect the results as they may not produce identical runoff amount and distribution.

Reply
We amended the statement to include the RCM model (and version) as well.

L396-399: Trends and variability are similar for Greenland land ice, tundra and non-Greenland land ice, i.e., although with a positive runoff offset in the current study. However, it is not the case for non- Greenland tundra. Could you elaborate on this?

Reply
We believe this could be related to the variety of model versions and re-analysis forcings used by Bamber et al. (2018). For Greenland they use RACMO2.3p2, while outside Greenland they use RACMO2.3p1 versions. Also, their RCMs are forced by ERA-40 between 1958-1978 and ERA-Interim between 1979-2016 (Noël et al. 2015 2017). For non-Greenland tundra their product is in better alignment with our results post ~1980 (roughly the start of ERA-Interim forcing).

We amended the text to reflect this.

L400-403: I am not sure to understand how statistical downscaling in tundra regions is important for Greenland (L392-395) but not for other Arctic regions? Is the tundra region rougher in Greenland? Please explain.

Reply
We agree that this argument is confusing.
We have completely removed it from the revised Section 5.3.

L407: I am confused, I understood that MAR at 6 km was used as input. Was it formerly downscaled to higher resolution before applying your downscaling technique? Please clarify.

Reply

*Yes, 6 km MAR (without prior downscaling) was used as the input. We edited the text to avoid misunderstandings.*

L411-413: I strongly recommend performing a thorough data evaluation, otherwise it is impossible to assess whether this new data set is robust, or an improvement upon other products.

*Reply*
*We have now included a validation against field river gauge observations and found that the predictive performance of our dataset is similar to the product of Mankoff et al. (2020). We therefore, consider our product an improvement in terms of spatial coverage (compared to Mankoff et al., 2020) and resolution (compared to Bamber et al., 2018).*

L415-420: The fact that improvement was found in Noël et al. (2016) using RACMO, with a different downscaling technique, does not imply similar results when using MAR as input. For instance, Tedesco et al. (2023) suggest that mass conservative statistical downscaling (i.e., no runoff increase) is required to better align with in-situ measurements in Greenland.

*Reply*
*We agree that these measurements of performance cannot be directly applied to our results. Not just because of the different input data and techniques, which have differences but are based around the same core idea, but also because these studies (i.e. Noël et al., 2016 and Tedesco et al. 2023) downscaled SMB, which they then validated against field observations (to determine formal uncertainty).*
*Carrying out the same exercise for runoff is difficult as localised field measurements of runoff are not available (the same way as SMB, e.g. the PROMICE dataset by Macguth et a., 2022). However, we think it is still informative to cite the results of these downscaling studies, together with the outcomes of our validation against river gauge data.*

*The text has been edited to better reflect these arguments, along with a detailed explanation about the which aspects of our dataset can be considered an improvement upon previous products (and in what way it is similar to them).*

L420-422: The variability and trends are mostly similar between products, but the runoff offset in this study remains important, calling for a proper model evaluation.

*Reply*
*We now include both a validation against field measurements, and a comparison of original and downscaled runoff. We also investigate the strongest factors that determine the effect of downscaling on the runoff. Based on this information, we argue that the predictive performance of our dataset is similar to previous products. The offsets are mainly attributed to inherent properties of our MAR inputs, which are modulated by the downscaling procedure.*

**Figures**

Fig. 2: Please add a scale bar for surface elevation.

*Reply*

A scale bar for surface elevation is now added to Figure 2.

**References**

Mankoff et al. (2020): https://essd.copernicus.org/articles/12/2811/2020/

Noël et al. (2016): https://tc.copernicus.org/articles/10/2361/2016/tc-10-2361-2016.html

Tedesco et al. (2023): https://tc.copernicus.org/articles/17/5061/2023/

Franco et al. (2012): https://tc.copernicus.org/articles/6/695/2012/

Bamber et al. (2018): https://iopscience.iop.org/article/10.1088/1748-9326/aac2f0/meta

Ryan et al. (2019): https://www.science.org/doi/10.1126/sciadv.aav3738

Reference:

Bamber et al. (2001)  https://doi.org/10.1029/2000JB900365

Fettweis et al. (2020) https://doi.org/10.5194/tc-14-3935-2020

Noël et al. (2023) https://doi.org/10.1038/s41467-023-43584-6

---

## Author Comment (AC2)

Review of

A high-resolution pan-Arctic meltwater discharge dataset from 1950 to 2021

by Igneczi and Bamber

General

This paper presents a new daily, 250 m resolution, 71-year runoff dataset for the Arctic, partitioning between runoff from ice and tundra. Its main strengths are the high temporal and spatial resolution, long time series and consistent source data treatment. Its main weaknesses are using only a single model product and the lack of detailed (regional) evaluation. This makes it hard to judge whether the (sometimes significant) differences that are found when comparing with previous products represent real improvements. See major comments below.

Reply
Thank you for your suggestions. We have undertaken additional data evaluation steps including a validation against river gauge data, and comparisons of the original and downscaled runoff. This has led to a significant revision of Section 5 and 6. We hope this will aid the evaluation of our manuscript.

Major comments

l. 117: If you go from 90 to 250 m resolution, how do you deal with fractional ice cover?

Reply
To keep data processing streamlined we relied on simple nearest neighbour interpolation for GIMP re-gridding, similarly a grid cell was considered ice covered if its centroid was within RGI ice cover polygons. This simple procedure circumvented the need to consider fractional coverages and allowed us to consolidate all data inputs to the same resolution and grid, which makes our steps less complex. Although ice masks are less precise due to this simplification and their lower resolution, we found that their total area remained within ±1% of the original.

Additional description is provided at the relevant sections to explain the data processing more precisely.

l. 141: Was the RCM forced hourly by ERA5? Usually this is every three hours. How do you assess non-glaciated runoff from regions not covered by the MAR domain?

Reply
Thank you for pointing this out, MAR was forced by 6 hourly ERA5 (we omitted the number by mistake).

We only consider the areas – within the RGI regions - that are covered by MAR. Fortunately, Greenland, Svalbard and Iceland are covered completely.

However, steps need to be taken to ensure data consistency and quality in the Arctic Canada North and South, and Russian Arctic regions, e.g. we only retain drainage basins that have at least 90% of their area within the MAR domain to limit the extrapolation of MAR outputs. As our study areas (i.e. MAR coverage within RGI regions) are predominantly on islands (e.g. Baffin, Ellesmere, Novaya Zemlya) major drainage basins (potentially transporting water from outside the MAR domain) were not removed by this step (please see that attached map and statistics). The drainage basins are provided with out discharge dataset, so users can precisely investigate the origin of the coastal drainage.

[Figure]

Dissolved drainage basins outline (blue) is shown for the regions affected by the issue of basins covering areas without MAR data. The most significant effect is visible in the Arctic Canada South region (e.g. on the northern tip of the Labrador peninsula).

The relative effect (of the removal of such basins) on total integrated area is
Greenland: 0%
Svalbard: 0%

Iceland: 0%
Canada North: +1.01%
Canada South: -3.85%
Russian Arctic: -2.69%

We now highlight this issue better in the text (starting in Section 3.2.) and point the readers to the relevant sections where we explain in detail how we deal with the situation (when delineating drainage basins in Section 4.1 and when comparing our results with previous studies in Section 5.3).

Section 4.1: Although I appreciate that the authors prefer consistency in their calculations, it would be good to show/discuss the potential impact of solely relying on surface routing over e.g. the Greenland ice sheet.

Reply
We agree with the reviewer that this issue is important to discuss.

We provide more explanation in Section 4.1 as to why we avoided using ice thickness data and routing based on subglacial pressure heads.

Please briefly discuss how findings in this recent paper, which shows that meltwater is stored in the glacial Greenland system for significant amounts of time, could affect your results:
https://www.nature.com/articles/s41586-024-08096-3

Reply
We now discuss the effects buffered water storage can have on our coastal discharge data in Section 6.

l. 260: What albedo product was used in MAR? Albedo does typically not vary smoothly, making it harder to downscale as a function of elevation. Allowing albedo only to increase with increasing elevation may not be a valid assumption in many areas. How are the albedo corrections used in the final runoff product?

Reply
Thank you for raising this important point.
MAR uses the CROCUS snow model formulations, which have their minimum albedo set to 0.7, and MODIS based empirical firn and bare ice albedo parameterisations (e.g Fettweis et al., 2013; 2017).
We acknowledge that albedo variability is especially complex, and just relying on vertical gradients will not provide a precise downscaled product. This is the principal reason why we are not using the downscaled albedo product in any way to correct the downscaled runoff (e.g. similar to MODIS data used by Noël et al. 2016); it only provides contextual information (i.e. to estimate the ratio of runoff originating from below the snowline) that some users might find useful. As we only aim to locate the snowline, we are mostly affected by the CROCUS snow model formulations within MAR, which have their minimum albedo set to 0.7 (e.g Fettweis et

al., 2013; 2017). Thus, we are less affected by bare ice albedo from MAR which is particularly uncertain (e.g. Ryan et al., 2019).

We have added a new sentence to Section 2 to state in advance that ice albedo is not used to adjust the downscaled runoff. Section 4.2 was also edited to make this clear.

Section 4: Although useful, a bulk evaluation does not necessarily align with the bulk of the applications and users, which may well predominantly use dingle basin timeseries.

Reply
We agree that more local comparisons are useful. Unfortunately, there are few data sources that allow such an evaluation. In order to try and achieve this we have compared our coastal discharge time-series with corresponding Greenlandic river gauge data, using previously published methodology (Mankoff et al., 2020). This provides information about the localised performance of our data product. To summarise, the performance of our dataset is very similar to the MAR derived product of Mankoff et al. (2020).

l. 374: "We propose...". This and later hypotheses can be -at least partly- confirmed or rejected by comparing the runoff products in elevation bins: is the difference indeed deriving from the lower elevations which are better resolved? Same for non-Greenland ice.

Reply
We have now carried out detailed comparisons between the original and downscaled runoff. We also investigated the topographical configuration of the RGI regions, i.e. histograms and differences between MAR and high-resolution DEMs. Using these insights we have added a new Section (5.2) and several figures in the Supplementary material that deal with this issue.

l. 392: I find it unlikely that resolution is the only/leading explanation for the large differences in Greenland tundra runoff. This can be relatively easily checked by comparing total tundra area, the depth of the seasonal snow cover and rainfall. This can also be used to provide a more robust answer to the question why tundra runoff outside Greenland agrees better (although the variability is again more different).

Reply
We agree that additional information and explanation is needed.
Additional information is now included (Section 5.2) that aids unraveling the origin of this significant difference (Section 5.3 is also significantly revised). Our downscaling procedure increases net tundra runoff by about 7.3% in Greenland, due to better representation of low lying unglaciated valleys (Figure 6, S3, S5). However, the majority of the difference can be attributed to inherent differences in our MAR input i.e. MAR v3.11.5 (this study), versus MAR v3.11 downscaled from 7.5 km to 1 km (Mankoff et al., 2020), and RACMO2.3p2 downscaled from 11 km to 1 km (Bamber et al., 2018).

Bamber et al. (2018) uses a variety of model versions and re-analysis forcings. For Greenland

they use RACMO2.3p2, while outside Greenland they use RACMO2.3p1 versions. Also, their RCMs are forced by ERA-40 between 1958-1978 and ERA-Interim between 1979-2016 (Noël et al. 2015 2017). Thus, comparing the alignment of our dataset with the Bamber et al. (2018) product across regions (i.e. Greenland vs. non-Greenland) is difficult. E.g. for non-Greenland tundra their product is in better alignment with our results post ~1980 (roughly the start of ERA-Interim forcing).

IN general, I miss a direct comparison between the non-downscaled and downscaled products. Where/when do the differences occur, and can it be objectively assessed whether the downscaling improves upon the original products? It presumably does, but unless it is somehow quantified this remains speculative.

Reply
We agree with the reviewer this such comparisons are necessary.
In the revision of the manuscript we include comparisons between our downscaled and original MAR runoff, separately for tundra and ice in each investigated RGI region. A new section is now included to explain the findings (Section 5.2). To summarise, bulk ice runoff slightly increases in Greenland due to downscaling (+2.4%) but decreases elsewhere (between -4.4 and -23.5%). Bulk tundra runoff increases due to downscaling in all regions (between 4.2 and 28%). We have also investigated the potential factors that could have influenced the net effect of downscaling on bulk runoff. We have found that differences in the MAR and high-resolution ice- and land masks, and DEMs, along with the topographical configuration of each region provide reasonable explanations.
We reviewed Section 5.3 given these new insights. Overall, we propose that the observed differences between this study and previous products are not primarily caused by our downscaling procedure, as they are mostly inherent to the MAR inputs.

Minor comments

l. 25: warmed -> increased (my strong preference!)

Reply
Done.

l. 159: This equation holds for runoff from land ice, please specify.

Reply
We added that there is no retention or refreezing for tundra runoff.

l. 340: If find the reasoning for distinguishing runoff from above and below the snow line hard to follow. Why is it relevant? Figure 6 suggests that the large majority of runoff comes from below the snow line. Interpretation?

Reply

Distinguishing between liquid discharge sourced directly from seasonal snow (i.e. above the snowline) and from firn/ice which represent a "reservoir" source could be useful for certain perturbation experiments (e.g. examining fjord circulation) that aim to pinpoint the specific effect of melting ice (while controlling for precipitation). We do not consider this as a primary output, but though it might be useful for some users.

Although the annual amount of runoff from above the snowline is small, it could be more significant early in the melt season (the snowline is tracked daily). Also, MAR is prone to the overestimation of bare ice area (Ryan et al., 2019; Fettweis et al., 2020), thus true snowlines might be located lower than our estimates. This is now pointed out prominently in Section 4.4.

l. 410: What is meant by "its overall uncertainty"? I presume you mean the uncertainty in runoff?

Reply
Yes, we have edited the text to make this more clear.

---

## Referee Report (RR1)

**Review of *"A high-resolution pan-Arctic meltwater discharge dataset from 1950 to 2021"***
by Adam Igneczi et al.

This is my second review of the manuscript by Igneczi et al. Overall, the authors well addressed my previous concerns, but some clarifications are still required. The authors can find my comments below.

**Response letter**
**Reviewer #1 L124:** I understand that the authors prefer using a binary mask retrieved from a nearest neighbor interpolation over creating a fractional mask. However, at high-resolution, binary masks may lead to large area discrepancies, notably for small glaciers and ice caps. Table 1 suggests relatively small area differences between original and downscaled MAR, but this does not imply that these areas compare well with GIMP/RGI reference masks. To address this, the authors could report in Table 1 the difference in integrated ice mask area between GIMP/RGI, and the original (5 km) and downscaled (250 m) MAR grids for each investigated region. See also my comment on Table 1 below.

**Point comments**
My comments are based on the line numbering of the tracked-change document.

L20-21: As mentioned in my previous review, and to avoid being misleading, I strongly recommend clarifying that daily runoff data are spatially integrated over relevant catchments/basins, i.e., not gridded at 250 m.

L71-77: Same comment here, it would be beneficial to clarify that the final data set is spatially integrated over catchments/basins.

L260-262: From Fig. 3, it looks like large areas are locally discarded from the study based on your 90% threshold. At least, it would be interesting to give (1) insight on the discarded area fraction compared to the total area, i.e., discarded area / total area (%); (2) briefly elaborate on why spatial extrapolation is not suitable over these discarded areas, e.g., are the neighboring estimated gradients not representative/suitable enough?

L307: "Ice and land runoff were handled separately." Please, briefly elaborate on why doing so is important as you did in your response letter, e.g., large runoff contrast at the ice/land interface. Same comment in L351-352.

L312-314: Please give an example of what you mean by "elevation independent variance" as in the response letter, e.g., firn retention processes nearby the equilibrium line.

L366-367: Are annual data gridded at 250 m? Are they part of the published data set? Please clarify.

L449: Could you explicitly write down your $R^2$ values in the text (Fig. 8) to facilitate interpretation/comparison. It would be good to list mean bias and RMSE (model vs. measurements) in Figs. 8a-b, and report the values in the main text.

L468-513: I do not think that using "over or underestimate" is correct when comparing downscaled and original MAR data, as this comparison does not involve observations. Please replace "overestimate" by "is larger than" and underestimate by "is smaller than" (or equivalent) where appropriate, e.g., L468, 471, 477, 492,499, and 510.

L479-482: "However, this is … also need to be considered." These sentences are unclear, please reformulate. Do you mean that ice area between downscaled and original MAR does not change much except for the Russian Arctic? This is surprising, especially when comparing the ice/tundra area difference at 5 km and 250 m in, e.g., Figs. 5a-c and 6a-c in Canada and Fig. S1a-c in Greenland.

Table 1: This is an interesting comparison; however, I miss the difference in ice mask area between reference GIMP/RGI ice masks, and those from downscaled and original MAR for all regions. I

recommend adding this information as additional columns in Table 1, and briefly report the outcome in the main text, e.g., near L479-481.

L582: What do you mean by "static", a fixed topography and ice mask in MAR? Please, clarify.

**Style**
L14-15: I suggest "To date, meltwater discharge data at Arctic coastlines are only available from two datasets that are limited by their spatial resolution and/or coverage."

L71-77: You could split this long sentence at L75 after "1950-2021" as "… for the period 1950-2021. Our database is publicly available, efficiently stored, and covers the most important …". Do you mean "publicly available" by "easily accessible"? Please clarify.

L286: "MAR variables within an 8-neighbourhood (8-N) moving window." And then in L304: "…, first, an 8-N moving window was applied …"

L418 and 420: I would recommend "5.1 Evaluation against …" and "To evaluate our product …".

L464: "specific" instead of "characteristic"?

L 477: "Lower ice runoff in downscaled MAR mostly stems from reduction in ice area …"

L597-598: "… is difficult to estimate as localized in-situ runoff measurements are extremely sparse."

L604: "… against in-situ measurements collected in the field,  and found that …"

L615: "… but not in terms of predictive performance …"

L641: "For instance, the duration of buffered …"

**Figures**
Fig. 9 caption in L476 "… for (a) ice and (b) land areas …"

**Supplement**
Fig. S2 caption: "However, it is important to note that the edges of the integrated basins were …", and "one third of the aggregated …"

Fig. S4 supplement text: What do you mean by "We propose that this is due to the topographical configuration of the ice coverage"? This is vague, please clarify.

Fig. S5: "COP-250 DEM is lower than MAR DEM towards lower elevations and vice versa." From the graphs, it looks like the opposite, i.e., positive values in lower areas (COP-250 > MAR) and negative values higher up (COP-250 < MAR). Please verify this carefully here, in Fig. S4 and in the main text when referring to these Figures.

Fig. S6 supplement text: In the last line of the paragraph, do you mean "more intensive runoff per unit area"?

---

## Author Response (AR2)

**Review of "A high-resolution pan-Arctic meltwater discharge dataset from 1950 to 2021"**

by Adam Igneczi et al.

This is my second review of the manuscript by Igneczi et al. Overall, the authors well addressed my previous concerns, but some clarifications are still required. The authors can find my comments below.

**Reply**

Thank you for the constructive review. Please find our responses below.

**Response letter**

**Reviewer #1 L124:** I understand that the authors prefer using a binary mask retrieved from a nearest neighbor interpolation over creating a fractional mask. However, at high-resolution, binary masks may lead to large area discrepancies, notably for small glaciers and ice caps. Table 1 suggests relatively small area differences between original and downscaled MAR, but this does not imply that these areas compare well with GIMP/RGI reference masks. To address this, the authors could report in Table 1 the difference in integrated ice mask area between GIMP/RGI, and the original (5 km) and downscaled (250 m) MAR grids for each investigated region. See also my comment on Table 1 below.

**Reply**

Thank you for this suggestion. We agree that including the reference masks (shapefiles and 90 m resolution rasters) in the comparisons will improve confidence. We have now carried out area comparisons between the original reference masks (RGI, GIMP, Copernicus) and the converted versions (i.e. resampled to 250 m) that were used for downscaling. Overall, we find that area discrepancies remain within the  $\pm 1\%$  range.

These discrepancies are small and would be hard to spot in Table 1. Also, they do not directly influence the downscaling procedure (i.e. as only the 250 m masks are used there, which we already compare with the MAR fractional masks).

Thus, we discuss these comparisons in Section 3.1.2, together with the data processing steps, to avoid overcomplicating Section 5.2. We have added a new table (Table 1) with all the results and discuss it in a few new sentences.

**Point comments**

My comments are based on the line numbering of the tracked-change document.

**L20-21:** As mentioned in my previous review, and to avoid being misleading, I strongly recommend clarifying that daily runoff data are spatially integrated over relevant catchments/basins, i.e., not gridded at 250 m.

**Reply**

Thank you for the suggestion. We agree that it is important to communicate this clearly. We have added the following clarification to the abstract: "..Coastal meltwater discharge data – i.e. spatially integrated runoff that is assigned to the outflow points of drainage basins..."

L71-77: Same comment here, it would be beneficial to clarify that the final data set is spatially integrated over catchments/basins.

**Reply**

We agree with this suggestion; we added the following amendments to the relevant section: "...algorithms to estimate coastal surface runoff fluxes by reporting spatially integrated runoff at coastal outflow points. Bamber..."

"...available, efficiently stored – i.e. by reporting runoff that is spatially integrated over drainage basins – and covers  $\dots$ "

**L260-262:** From Fig. 3, it looks like large areas are locally discarded from the study based on your 90% threshold. At least, it would be interesting to give (1) insight on the discarded area fraction compared to the total area, i.e., discarded area / total area (%); (2) briefly elaborate on why spatial extrapolation is not suitable over these discarded areas, e.g., are the neighboring estimated gradients not representative/suitable enough?

**Reply**

The footprint of Figure 3 was chosen specifically to illustrate the effect of this step (i.e. it is one of the most strongly affected area in the Arctic).

Although spatial extrapolation is permitted (and we think it is appropriate over small distances), we wanted to strongly curtail extrapolation beyond MAR domains to provide a more robust dataset (hence the 90% threshold) and err on the side of caution. Though we concede that a lower threshold would also be appropriate.

We added the following section to the text to aid the readers in assessing the scale of this step. "...Thus, altogether, 1.01%, 2.68%, and 3.85% of the MAR domain was discarded in Arctic Canada North, Russian Arctic, Arctic Canada South, respectively. Other regions were unaffected by this step, and the discarded area had negligible ice coverage."

**L307:** "Ice and land runoff were handled separately." Please, briefly elaborate on why doing so is important as you did in your response letter, e.g., large runoff contrast at the ice/land interface. Same comment in L351-352.

**Reply**

We amended the sentence to indicate the importance of considering the large runoff contract at the ice-tundra interface: "Ice and land runoff were handled separately to prevent "leakage" due to large runoff contrast at the ice-tundra interface."

"....Similar to the calculation of the vertical gradients, ice and tundra runoff were handled separately to prevent biases caused by the high runoff contrast at the ice-tundra interface. Henceforth..."

**L312-314:** Please give an example of what you mean by "elevation independent variance" as in the response letter, e.g., firn retention processes nearby the equilibrium line.

**Reply**

We have included a few examples at the relevant section.

**L366-367:** Are annual data gridded at 250 m? Are they part of the published data set? Please clarify.

**Reply**

Yes, but due to their large size (~60 GB altogether) they are not part of the published data set that we've uploaded to Pangea. We added the following clarification: "...Due to their large size, these files are not published, but they are available on request..."

**L449:** Could you explicitly write down your R2 values in the text (Fig. 8) to facilitate interpretation/comparison. It would be good to list mean bias and RMSE (model vs. measurements) in Figs. 8a-b, and report the values in the main text.

**Reply**

We now include RMSE and MBE on the figure. Also, we quote our own  $R^2$  in the text to facilitate comparisons.

**L468-513:** I do not think that using "over or underestimate" is correct when comparing downscaled and original MAR data, as this comparison does not involve observations. Please replace "overestimate" by "is larger than" and underestimate by "is smaller than" (or equivalent) where appropriate, e.g., L468, 471, 477, 492,499, and 510.

**Reply**

We agree that over- and underestimation are not the precise terms to use here. We have revised this section to use more appropriate terminology.

L479-482: "However, this is ... also need to be considered." These sentences are unclear, please reformulate. Do you mean that ice area between downscaled and original MAR does not change much except for the Russian Arctic? This is surprising, especially when comparing the ice/tundra area difference at 5 km and 250 m in, e.g., Figs. 5a-c and 6a-c in Canada and Fig. S1a-c in Greenland.

**Reply**

Thank you pointing out these confusing sentences.

Ice area does in fact change elsewhere, even more so than in the Russian Arctic (see Table 1). What we were trying to point out is that the change in tundra area is not reciprocal to the change in land area (x % decrease in ice area will not lead to x % increase in tundra area), and in some regions (e.g. in the Russin Arctic) tundra area decreases (slightly) while tundra runoff increases. Thus, topography needs to be considered.

Please note that ice and land runoff is shown on Fig 5a and Fig 6a where the fractional MAR mask indicates some ice/land coverage (which could be just a few % of the pixel), thus these figures should not be used to compare ice/tundra area differences. We have amended the caption to warn readers about this.

**Table 1:** This is an interesting comparison; however, I miss the difference in ice mask area between reference GIMP/RGI ice masks, and those from downscaled and original MAR for all regions. I recommend adding this information as additional columns in Table 1, and briefly report the outcome in the main text, e.g., near L479-481.

**Reply**

We agree that further comparisons are useful (please see our previous reply). Overall, we find that area discrepancies between the original masks (shapefiles and 90 m rasters) and their 250 m versions are small, remaining within the  $\pm 1\%$  range.

As these discrepancies are small, they would be hard to spot in Table 1. Thus, we discuss them explicitly in Section 3.1.2.

**L582:** What do you mean by "static", a fixed topography and ice mask in MAR? Please, clarify.

**Reply**

Yes, we mean time-independent forcing (e.g. a fixed ice mask). We have added examples to clarify.

**Style**

**L14-15:** I suggest "To date, meltwater discharge data at Arctic coastlines are only available from two datasets that are limited by their spatial resolution and/or coverage."

Reply Corrected.

L71-77: You could split this long sentence at L75 after "1950-2021" as "... for the period 1950-2021. Our database is publicly available, efficiently stored, and covers the most important ...". Do you mean "publicly available" by "easily accessible"? Please clarify.

Reply

We have split the sentence. Yes, we mean publicly available.

**L286:** "MAR variables within an 8-neighbourhood (8-N) moving window." And then in L304: "..., first, an 8-N moving window was applied ..."

Reply

Thank you for noticing this. We have corrected it.

L418 and 420: I would recommend "5.1 Evaluation against …" and "To evaluate our product …".

Reply Corrected.

L464: "specific" instead of "characteristic"?

Reply

Yes, specific is a better choice. We have switched the word.

L 477: "Lower ice runoff in downscaled MAR mostly stems from reduction in ice area ..."

Reply We agree that this read better. We have edited the relevant section.

**L597-598:** "... is difficult to estimate as localized in-situ runoff measurements are extremely sparse."

Reply

Thank you for the suggestion. We have edited the relevant sentence.

L604: "... against in-situ measurements collected in the field, and found that ..."

Reply Corrected.

L615: "... but not in terms of predictive performance ..."

Reply Corrected.

L641: "For instance, the duration of buffered ..."

Reply Corrected.

**Figures**

Fig. 9 caption in L476 "... for (a) ice and (b) land areas ..."

Reply Thank you for spotting this. Corrected.

**Supplement**

**Fig. S2** caption: "However, it is important to note that the edges of the integrated basins were ...", and "one third of the aggregated ..."

Reply We revised this section.

**Fig. S4** supplement text: What do you mean by "We propose that this is due to the topographical configuration of the ice coverage"? This is vague, please clarify.

Reply

We added a short explanation to briefly outline the idea before discussing the details "..i.e. the predominant type and geometry of ice bodies (e.g. valley glaciers versus ice caps)..."

**Fig. S5:** "COP-250 DEM is lower than MAR DEM towards lower elevations and vice versa." From the graphs, it looks like the opposite, i.e., positive values in lower areas (COP-250 >

MAR) and negative values higher up (COP-250 < MAR). Please verify this carefully here, in Fig. S4 and in the main text when referring to these Figures.

Reply

We revised the section to better reflect the figure. Previously we were referring to Svalbard and the Russian Arctic (where COP-250 DEM is lower than MAR at low elevations). Now we include a more comprehensive discussion to cover regions where the situation mentioned in the comment is more typical (e.g. Canada North, Canada South).

**Fig. S6** supplement text: In the last line of the paragraph, do you mean "more intensive runoff per unit area"?

Reply Yes, we corrected this.